# Conservative Q-Learning
# for Offline Reinforcement Learning

**Aviral Kumar[1], Aurick Zhou[1], George Tucker[2], Sergey Levine[1,2]**
[1]UC Berkeley, [2]Google Research, Brain Team
`aviralk@berkeley.edu`

## Abstract

Effectively leveraging large, previously collected datasets in reinforcement learning (RL) is a key challenge for large-scale real-world applications. Offline RL algorithms promise to learn effective policies from previously-collected, static datasets without further interaction. However, in practice, offline RL presents a major challenge, and standard off-policy RL methods can fail due to overestimation of values induced by the distributional shift between the dataset and the learned policy, especially when training on complex and multi-modal data distributions. In this paper, we propose *conservative Q-learning (CQL)*, which aims to address these limitations by learning a conservative Q-function such that the expected value of a policy under this Q-function lower-bounds its true value. We theoretically show that CQL produces a lower bound on the value of the current policy and that it can be incorporated into a policy learning procedure with theoretical improvement guarantees. In practice, CQL augments the standard Bellman error objective with a simple Q-value regularizer which is straightforward to implement on top of existing deep Q-learning and actor-critic implementations. On both discrete and continuous control domains, we show that CQL substantially outperforms existing offline RL methods, often learning policies that attain 2-5 times higher final return, especially when learning from complex and multi-modal data distributions.

## 1 Introduction

Recent advances in reinforcement learning (RL), especially when combined with expressive deep network function approximators, have produced promising results in domains ranging from robotics [29] to strategy games [4] and recommendation systems [35]. However, applying RL to real-world problems consistently poses practical challenges: in contrast to the kinds of data-driven methods that have been successful in supervised learning [22, 10], RL is classically regarded as an active learning process, where each training run requires active interaction with the environment. Interaction with the real world can be costly and dangerous, and the quantities of data that can be gathered online are substantially lower than the offline datasets that are used in supervised learning [9], which only need to be collected once. Offline RL, also known as batch RL, offers an appealing alternative [11, 15, 30, 3, 27, 54, 34]. Offline RL algorithms learn from large, previously collected datasets, without interaction. This in principle can make it possible to leverage large datasets, but in practice fully offline RL methods pose major technical difficulties, stemming from the distributional shift between the policy that collected the data and the learned policy. This has made current results fall short of the full promise of such methods.

Directly utilizing existing value-based off-policy RL algorithms in an offline setting generally results in poor performance, due to issues with bootstrapping from out-of-distribution actions [30, 15] and overfitting [13, 30, 3]. This typically manifests as erroneously optimistic value function estimates. If we can instead learn a *conservative* estimate of the value function, which provides a lower bound on the true values, this overestimation problem could be addressed. In fact, because policy evaluation

and improvement typically only use the value of the policy, we can learn a less conservative lower bound Q-function, such that only the expected value of Q-function under the policy is lower-bounded, as opposed to a point-wise lower bound. We propose a novel method for learning such conservative Q-functions via a simple modification to standard value-based RL algorithms. The key idea behind our method is to minimize values under an appropriately chosen distribution over state-action tuples, and then further tighten this bound by also incorporating a *maximization* term over the data distribution.

Our primary contribution is an algorithmic framework, which we call conservative Q-learning (CQL), for learning conservative, lower-bound estimates of the value function, by regularizing the Q-values during training. Our theoretical analysis of CQL shows that *only* the expected value of this Q-function under the policy lower-bounds the true policy value, preventing extra under-estimation that can arise with point-wise lower-bounded Q-functions, that have typically been explored in the opposite context in exploration literature [46, 26]. We also empirically demonstrate the robustness of our approach to Q-function estimation error. Our practical algorithm uses these conservative estimates for policy evaluation and offline RL. CQL can be implemented with less than **20** lines of code on top of a number of standard, online RL algorithms [19, 8], simply by adding the CQL regularization terms to the Q-function update. In our experiments, we demonstrate the efficacy of CQL for offline RL, in domains with complex dataset compositions, where prior methods are typically known to perform poorly [12] and domains with high-dimensional visual inputs [5, 3]. CQL outperforms prior methods by as much as **2-5x** on many benchmark tasks, and is the only method that can outperform simple behavioral cloning on a number of realistic datasets collected from human interaction.

## 2 Preliminaries

The goal in reinforcement learning is to learn a policy that maximizes the expected cumulative discounted reward in a Markov decision process (MDP), which is defined by a tuple $(\mathcal{S}, \mathcal{A}, T, r, \gamma)$. $\mathcal{S}, \mathcal{A}$ represent state and action spaces, $T(\mathbf{s}'|\mathbf{s}, \mathbf{a})$ and $r(\mathbf{s}, \mathbf{a})$ represent the dynamics and reward function, and $\gamma \in (0, 1)$ represents the discount factor. $\pi_\beta(\mathbf{a}|\mathbf{s})$ represents the behavior policy, $\mathcal{D}$ is the dataset, and $d^{\pi_\beta}(\mathbf{s})$ is the discounted marginal state-distribution of $\pi_\beta(\mathbf{a}|\mathbf{s})$. The dataset $\mathcal{D}$ is sampled from $d^{\pi_\beta}(\mathbf{s})\pi_\beta(\mathbf{a}|\mathbf{s})$. On all states $\mathbf{s} \in \mathcal{D}$, let $\hat{\pi}_\beta(\mathbf{a}|\mathbf{s}) := \frac{\sum_{\mathbf{s}, \mathbf{a} \in \mathcal{D}} \mathbf{1}[\mathbf{s}=\mathbf{s}, \mathbf{a}=\mathbf{a}]}{\sum_{\mathbf{s} \in \mathcal{D}} \mathbf{1}[\mathbf{s}=\mathbf{s}]}$ denote the empirical behavior policy, at that state. We assume that the rewards $r$ satisfy: $|r(\mathbf{s}, \mathbf{a})| \leq R_{\max}$.

Off-policy RL algorithms based on dynamic programming maintain a parametric Q-function $Q_\theta(s, a)$ and, optionally, a parametric policy, $\pi_\phi(a|s)$. Q-learning methods train the Q-function by iteratively applying the Bellman optimality operator $\mathcal{B}^* Q(\mathbf{s}, \mathbf{a}) = r(\mathbf{s}, \mathbf{a}) + \gamma \mathbb{E}_{\mathbf{s}' \sim P(\mathbf{s}'|\mathbf{s}, \mathbf{a})}[\max_{\mathbf{a}'} Q(\mathbf{s}', \mathbf{a}')]$, and use exact or an approximate maximization scheme, such as CEM [29] to recover the greedy policy. In an actor-critic algorithm, a separate policy is trained to maximize the Q-value. Actor-critic methods alternate between computing $Q^\pi$ via (partial) policy evaluation, by iterating the Bellman operator, $\mathcal{B}^\pi Q = r + \gamma P^\pi Q$, where $P^\pi$ is the transition matrix coupled with the policy: $P^\pi Q(\mathbf{s}, \mathbf{a}) = \mathbb{E}_{\mathbf{s}' \sim T(\mathbf{s}'|\mathbf{s}, \mathbf{a}), \mathbf{a}' \sim \pi(\mathbf{a}'|\mathbf{s}')}[Q(\mathbf{s}', \mathbf{a}')]$, and improving the policy $\pi(\mathbf{a}|\mathbf{s})$ by updating it towards actions that maximize the expected Q-value. Since $\mathcal{D}$ typically does not contain all possible transitions $(\mathbf{s}, \mathbf{a}, \mathbf{s}')$, the policy evaluation step actually uses an empirical Bellman operator that only backs up a single sample. We denote this operator $\hat{\mathcal{B}}^\pi$. Given a dataset $\mathcal{D} = \{(\mathbf{s}, \mathbf{a}, r\mathbf{s}')\}$ of tuples from trajectories collected under a behavior policy $\pi_\beta$:

$$\hat{Q}^{k+1} \leftarrow \arg\min_Q \mathbb{E}_{\mathbf{s}, \mathbf{a}, \mathbf{s}' \sim \mathcal{D}} \left[ \left( (r(\mathbf{s}, \mathbf{a}) + \gamma \mathbb{E}_{\mathbf{a}' \sim \hat{\pi}^k(\mathbf{a}'|\mathbf{s}')}[\hat{Q}^k(\mathbf{s}', \mathbf{a}')]) - Q(\mathbf{s}, \mathbf{a}) \right)^2 \right] \text{ (policy evaluation)}$$

$$\hat{\pi}^{k+1} \leftarrow \arg\max_\pi \mathbb{E}_{\mathbf{s} \sim \mathcal{D}, \mathbf{a} \sim \pi^k(\mathbf{a}|\mathbf{s})} \left[ \hat{Q}^{k+1}(\mathbf{s}, \mathbf{a}) \right] \quad \text{(policy improvement)}$$

Offline RL algorithms based on this basic recipe suffer from action distribution shift [30, 59, 27, 34] during training, because the target values for Bellman backups in policy evaluation use actions sampled from the learned policy, $\pi^k$, but the Q-function is trained only on actions sampled from the behavior policy that produced the dataset $\mathcal{D}$, $\pi_\beta$. Since $\pi$ is trained to maximize Q-values, it may be biased towards out-of-distribution (OOD) actions with erroneously high Q-values. In standard RL, such errors can be corrected by attempting an action in the environment and observing its actual value. However, the inability to interact with the environment makes it challenging to deal with Q-values for OOD actions in offline RL. Typical offline RL methods [30, 27, 59, 54] mitigate this problem by constraining the learned policy [34] away from OOD actions. Note that Q-function training in offline RL does not suffer from state distribution shift, as the Bellman backup never queries the Q-function on out-of-distribution states. However, the policy may suffer from state distribution shift at test time.

# 3 The Conservative Q-Learning (CQL) Framework

In this section, we develop a conservative Q-learning algorithm, such that the expected value of a policy under the learned Q-function lower-bounds its true value. Lower-bounded Q-values prevent the over-estimation that is common in offline RL settings due to OOD actions and function approximation error [34, 30]. We use the term CQL to refer broadly to both Q-learning and actor-critic methods. We start with the policy evaluation step in CQL, which can be used by itself as an off-policy evaluation procedure, or integrated into a complete offline RL algorithm, as we will discuss in Section 3.2.

## 3.1 Conservative Off-Policy Evaluation

We aim to estimate the value $V^\pi(\mathbf{s})$ of a target policy $\pi$ given access to a dataset, $\mathcal{D}$, generated by following a behavior policy $\pi_\beta(\mathbf{a}|\mathbf{s})$. Because we are interested in preventing overestimation of the policy value, we learn a *conservative*, lower-bound Q-function by additionally minimizing Q-values alongside a standard Bellman error objective. Our choice of penalty is to minimize the expected Q-value under a particular distribution of state-action pairs, $\mu(\mathbf{s}, \mathbf{a})$. Since standard Q-function training does not query the Q-function value at unobserved states, but queries the Q-function at unseen actions, we restrict $\mu$ to match the state-marginal in the dataset, such that $\mu(\mathbf{s}, \mathbf{a}) = d^{\pi_\beta}(\mathbf{s})\mu(\mathbf{a}|\mathbf{s})$. This gives rise to the iterative update for training the Q-function, as a function of a tradeoff factor $\alpha \geq 0$:

$$\hat{Q}^{k+1} \leftarrow \arg\min_Q \ \alpha \, \mathbb{E}_{\mathbf{s}\sim\mathcal{D}, \mathbf{a}\sim\mu(\mathbf{a}|\mathbf{s})}\left[Q(\mathbf{s}, \mathbf{a})\right] + \frac{1}{2}\mathbb{E}_{\mathbf{s},\mathbf{a}\sim\mathcal{D}}\left[\left(Q(\mathbf{s}, \mathbf{a}) - \hat{\mathcal{B}}^\pi \hat{Q}^k(\mathbf{s}, \mathbf{a})\right)^2\right], \quad (1)$$

In Theorem 3.1, we show that the resulting Q-function, $\hat{Q}^\pi := \lim_{k\to\infty} \hat{Q}^k$, lower-bounds $Q^\pi$ at all $\mathbf{s} \in \mathcal{D}, \mathbf{a} \in \mathcal{A}$. However, we can substantially tighten this bound if we are *only* interested in estimating $V^\pi(\mathbf{s})$. If we only require that the expected value of the $\hat{Q}^\pi$ under $\pi(\mathbf{a}|\mathbf{s})$ lower-bound $V^\pi$, we can improve the bound by introducing an additional Q-value *maximization* term under the data distribution, $\pi_\beta(\mathbf{a}|\mathbf{s})$, resulting in the iterative update (changes from Equation 1 in red):

$$\hat{Q}^{k+1} \leftarrow \arg\min_Q \ \alpha \cdot \left(\mathbb{E}_{\mathbf{s}\sim\mathcal{D}, \mathbf{a}\sim\mu(\mathbf{a}|\mathbf{s})}\left[Q(\mathbf{s}, \mathbf{a})\right] - \mathbb{E}_{\mathbf{s}\sim\mathcal{D}, \mathbf{a}\sim\hat{\pi}_\beta(\mathbf{a}|\mathbf{s})}\left[Q(\mathbf{s}, \mathbf{a})\right]\right)$$
$$+ \frac{1}{2}\mathbb{E}_{\mathbf{s},\mathbf{a},\mathbf{s}'\sim\mathcal{D}}\left[\left(Q(\mathbf{s}, \mathbf{a}) - \hat{\mathcal{B}}^\pi \hat{Q}^k(\mathbf{s}, \mathbf{a})\right)^2\right]. \quad (2)$$

In Theorem 3.2, we show that, while the resulting Q-value $\hat{Q}^\pi$ may not be a point-wise lower-bound, we have $\mathbb{E}_{\pi(\mathbf{a}|\mathbf{s})}[\hat{Q}^\pi(\mathbf{s}, \mathbf{a})] \leq V^\pi(\mathbf{s})$ when $\mu(\mathbf{a}|\mathbf{s}) = \pi(\mathbf{a}|\mathbf{s})$. Intuitively, since Equation 2 maximizes Q-values under the behavior policy $\hat{\pi}_\beta$, Q-values for actions that are likely under $\hat{\pi}_\beta$ might be overestimated, and hence $\hat{Q}^\pi$ may not lower-bound $Q^\pi$ pointwise. While in principle the maximization term can utilize other distributions besides $\hat{\pi}_\beta(\mathbf{a}|\mathbf{s})$, we prove in Appendix D.2 that the resulting value is not guaranteed to be a lower bound for other distribution besides $\hat{\pi}_\beta(\mathbf{a}|\mathbf{s})$, though other distributions besides $\hat{\pi}_\beta(\mathbf{a}|\mathbf{s})$ can still yield a lower bound if the Bellman error is also re-weighted to come from the distribution chosen to maximize the expected Q-value.

**Theoretical analysis.** We first note that Equations 1 and 2 use the empirical Bellman operator, $\hat{\mathcal{B}}^\pi$, instead of the actual Bellman operator, $\mathcal{B}^\pi$. Following [47, 26, 45], we use concentration properties of $\hat{\mathcal{B}}^\pi$ to control this error. Formally, for all $\mathbf{s}, \mathbf{a} \in \mathcal{D}$, with probability $\geq 1 - \delta$, $|\hat{\mathcal{B}}^\pi - \mathcal{B}^\pi|(\mathbf{s}, \mathbf{a}) \leq \frac{C_{r,T,\delta}}{\sqrt{|\mathcal{D}(\mathbf{s},\mathbf{a})|}}$, where $C_{r,T,\delta}$ is a constant dependent on the concentration properties (variance) of $r(\mathbf{s}, \mathbf{a})$ and $T(\mathbf{s}'|\mathbf{s}, \mathbf{a})$, and $\delta \in (0, 1)$ (see Appendix D.3 for details). For simplicity, we assume that $\hat{\pi}_\beta(\mathbf{a}|\mathbf{s}) > 0, \forall \mathbf{a} \in \mathcal{A}, \forall \mathbf{s} \in \mathcal{D}$. Let $\frac{1}{\sqrt{|\mathcal{D}|}}$ denote a vector of size $|\mathcal{S}||\mathcal{A}|$ containing square root inverse counts for each state-action pair, except when $\mathcal{D}(\mathbf{s}, \mathbf{a}) = 0$, in which case the corresponding entry is a very large but finite value $\delta \geq \frac{2R_{\max}}{1-\gamma}$. Now, we show that the conservative Q-function learned by iterating Equation 1 lower-bounds the true Q-function. Proofs can be found in Appendix C.

**Theorem 3.1.** *For any $\mu(\mathbf{a}|\mathbf{s})$ with $\operatorname{supp}\mu \subset \operatorname{supp}\hat{\pi}_\beta$, with probability $\geq 1 - \delta$, $\hat{Q}^\pi$ (the Q-function obtained by iterating Equation 1) satisfies:*

$$\forall \mathbf{s} \in \mathcal{D}, \mathbf{a}, \ \hat{Q}^\pi(s,a) \leq Q^\pi(\mathbf{s}, \mathbf{a}) - \alpha\left[(I - \gamma P^\pi)^{-1}\frac{\mu}{\hat{\pi}_\beta}\right](\mathbf{s}, \mathbf{a}) + \left[(I - \gamma P^\pi)^{-1}\frac{C_{r,T,\delta}R_{\max}}{(1-\gamma)\sqrt{|\mathcal{D}|}}\right](\mathbf{s}, \mathbf{a}).$$

*Thus, if $\alpha$ is sufficiently large, then $\hat{Q}^\pi(\mathbf{s}, \mathbf{a}) \leq Q^\pi(\mathbf{s}, \mathbf{a}), \forall \mathbf{s} \in \mathcal{D}, \mathbf{a}$. When $\hat{\mathcal{B}}^\pi = \mathcal{B}^\pi$, any $\alpha > 0$ guarantees $\hat{Q}^\pi(\mathbf{s}, \mathbf{a}) \leq Q^\pi(\mathbf{s}, \mathbf{a}), \forall \mathbf{s} \in \mathcal{D}, \mathbf{a} \in \mathcal{A}$.*

Next, we show that Equation 2 lower-bounds the expected value under the policy $\pi$, when $\mu = \pi$. We also show that Equation 2 does not lower-bound the Q-value estimates pointwise. For this result, we abuse notation and assume that $\frac{1}{\sqrt{|\mathcal{D}|}}$ refers to a vector of inverse square root of only state counts, with a similar correction as before used to handle the entries of this vector at states with zero counts.

**Theorem 3.2** (Equation 2 results in a tighter lower bound). *The value of the policy under the Q-function from Equation 2, $\hat{V}^\pi(\mathbf{s}) = \mathbb{E}_{\pi(\mathbf{a}|\mathbf{s})}[\hat{Q}^\pi(\mathbf{s}, \mathbf{a})]$, lower-bounds the true value of the policy obtained via exact policy evaluation, $V^\pi(\mathbf{s}) = \mathbb{E}_{\pi(\mathbf{a}|\mathbf{s})}[Q^\pi(\mathbf{s}, \mathbf{a})]$, when $\mu = \pi$, according to:*

$$\forall \mathbf{s} \in \mathcal{D}, \ \hat{V}^\pi(\mathbf{s}) \le V^\pi(\mathbf{s}) - \alpha \left[(I - \gamma P^\pi)^{-1} \mathbb{E}_\pi \left[\frac{\pi}{\hat{\pi}_\beta} - 1\right]\right](\mathbf{s}) + \left[(I - \gamma P^\pi)^{-1} \frac{C_{r,T,\delta} R_{\max}}{(1-\gamma)\sqrt{|\mathcal{D}|}}\right](\mathbf{s}).$$

*Thus, if $\alpha > \frac{C_{r,T} R_{\max}}{1-\gamma} \cdot \max_{\mathbf{s}\in\mathcal{D}} \frac{1}{|\sqrt{|\mathcal{D}(\mathbf{s})|}} \cdot \left[\sum_{\mathbf{a}} \pi(\mathbf{a}|\mathbf{s})(\frac{\pi(\mathbf{a}|\mathbf{s})}{\hat{\pi}_\beta(\mathbf{a}|\mathbf{s})} - 1)\right]^{-1}$, $\forall \mathbf{s} \in \mathcal{D}, \hat{V}^\pi(\mathbf{s}) \le V^\pi(\mathbf{s})$, with probability $\ge 1 - \delta$. When $\hat{\mathcal{B}}^\pi = \mathcal{B}^\pi$, then any $\alpha > 0$ guarantees $\hat{V}^\pi(\mathbf{s}) \le V^\pi(\mathbf{s}), \forall \mathbf{s} \in \mathcal{D}$.*

The analysis presented above assumes that no function approximation is used in the Q-function, meaning that each iterate can be represented exactly. We can further generalize the result in Theorem 3.2 to the case of both linear function approximators and non-linear neural network function approximators, where the latter builds on the neural tangent kernel (NTK) framework [25]. Due to space constraints, we present these results in Theorem D.1 and Theorem D.2 in Appendix D.1.

**In summary**, we showed that the basic CQL evaluation in Equation 1 learns a Q-function that lower-bounds the true Q-function $Q^\pi$, and the evaluation in Equation 2 provides a *tighter* lower bound on the expected Q-value of the policy $\pi$. For suitable $\alpha$, both bounds hold under sampling error and function approximation. We also note that as more data becomes available and $|\mathcal{D}(\mathbf{s}, \mathbf{a})|$ increases, the theoretical value of $\alpha$ that is needed to guarantee a lower bound decreases, which indicates that in the limit of infinite data, a lower bound can be obtained by using extremely small values of $\alpha$. Next, we will extend on this result into a complete RL algorithm.

### 3.2 Conservative Q-Learning for Offline RL

We now present a general approach for offline policy learning, which we refer to as conservative Q-learning (CQL). As discussed in Section 3.1, we can obtain Q-values that lower-bound the value of a policy $\pi$ by solving Equation 2 with $\mu = \pi$. How should we utilize this for policy optimization? We could alternate between performing full off-policy evaluation for each policy iterate, $\hat{\pi}^k$, and one step of policy improvement. However, this can be computationally expensive. Alternatively, since the policy $\hat{\pi}^k$ is typically derived from the Q-function, we could instead choose $\mu(\mathbf{a}|\mathbf{s})$ to approximate the policy that would maximize the current Q-function iterate, thus giving rise to an online algorithm.

We can formally capture such online algorithms by defining a family of optimization problems over $\mu(\mathbf{a}|\mathbf{s})$, presented below, with modifications from Equation 2 marked in red. An instance of this family is denoted by CQL($\mathcal{R}$) and is characterized by a particular choice of regularizer $\mathcal{R}(\mu)$:

$$\min_Q \max_\mu \ \alpha \left(\mathbb{E}_{\mathbf{s}\sim\mathcal{D}, \mathbf{a}\sim\mu(\mathbf{a}|\mathbf{s})}[Q(\mathbf{s}, \mathbf{a})] - \mathbb{E}_{\mathbf{s}\sim\mathcal{D}, \mathbf{a}\sim\hat{\pi}_\beta(\mathbf{a}|\mathbf{s})}[Q(\mathbf{s}, \mathbf{a})]\right)$$
$$+ \frac{1}{2} \mathbb{E}_{\mathbf{s}, \mathbf{a}, \mathbf{s}'\sim\mathcal{D}}\left[\left(Q(\mathbf{s}, \mathbf{a}) - \hat{\mathcal{B}}^{\pi_k}\hat{Q}^k(\mathbf{s}, \mathbf{a})\right)^2\right] + \mathcal{R}(\mu) \quad (\text{CQL}(\mathcal{R})). \quad (3)$$

**Variants of CQL.** To demonstrate the generality of the CQL family of objectives, we discuss two specific instances within this family that are of special interest, and we evaluate them empirically in Section 6. If we choose $\mathcal{R}(\mu)$ to be the KL-divergence against a prior distribution, $\rho(\mathbf{a}|\mathbf{s})$, i.e., $\mathcal{R}(\mu) = -D_{\text{KL}}(\mu, \rho)$, then we get $\mu(\mathbf{a}|\mathbf{s}) \propto \rho(\mathbf{a}|\mathbf{s}) \cdot \exp(Q(\mathbf{s}, \mathbf{a}))$ (for a derivation, see Appendix A). Frist, if $\rho = \text{Unif}(\mathbf{a})$, then the first term in Equation 3 corresponds to a soft-maximum of the Q-values at any state $\mathbf{s}$ and gives rise to the following variant of Equation 3, called CQL($\mathcal{H}$):

$$\min_Q \ \alpha \mathbb{E}_{\mathbf{s}\sim\mathcal{D}}\left[\log\sum_{\mathbf{a}}\exp(Q(\mathbf{s}, \mathbf{a})) - \mathbb{E}_{\mathbf{a}\sim\hat{\pi}_\beta(\mathbf{a}|\mathbf{s})}[Q(\mathbf{s}, \mathbf{a})]\right] + \frac{1}{2}\mathbb{E}_{\mathbf{s}, \mathbf{a}, \mathbf{s}'\sim\mathcal{D}}\left[\left(Q - \hat{\mathcal{B}}^{\pi_k}\hat{Q}^k\right)^2\right]. \quad (4)$$

Second, if $\rho(\mathbf{a}|\mathbf{s})$ is chosen to be the previous policy $\hat{\pi}^{k-1}$, the first term in Equation 4 is replaced by an exponential weighted average of Q-values of actions from the chosen $\hat{\pi}^{k-1}(\mathbf{a}|\mathbf{s})$. Empirically, we

find that this variant can be more stable with high-dimensional action spaces (e.g., Table 2) where it is challenging to estimate $\log \sum_{\mathbf{a}} \exp$ via sampling due to high variance. In Appendix A, we discuss an additional variant of CQL, drawing connections to distributionally robust optimization [43]. We will discuss a practical instantiation of a CQL deep RL algorithm in Section 4. CQL can be instantiated as either a Q-learning algorithm (with $\mathcal{B}^*$ instead of $\mathcal{B}^\pi$ in Equations 3, 4) or as an actor-critic algorithm.

**Theoretical analysis of CQL.** Next, we will theoretically analyze CQL to show that the policy updates derived in this way are indeed "conservative", in the sense that each successive policy iterate is optimized against a lower bound on its value. For clarity, we state the results in the absence of finite-sample error, in this section, but sampling error can be incorporated in the same way as Theorems 3.1 and 3.2, and we discuss this in Appendix C. Theorem 3.3 shows that CQL($\mathcal{H}$) learns Q-value estimates that lower-bound the actual Q-function under the action-distribution defined by the policy, $\pi^k$, under mild regularity conditions (slow updates on the policy).

**Theorem 3.3** (CQL learns lower-bounded Q-values). *Let $\pi_{\hat{Q}^k}(\mathbf{a}|\mathbf{s}) \propto \exp(\hat{Q}^k(\mathbf{s}, \mathbf{a}))$ and assume that $D_{\mathrm{TV}}(\hat{\pi}^{k+1}, \pi_{\hat{Q}^k}) \leq \varepsilon$ (i.e., $\hat{\pi}^{k+1}$ changes slowly w.r.t to $\hat{Q}^k$). Then, the policy value under $\hat{Q}^k$, lower-bounds the actual policy value, $\hat{V}^{k+1}(\mathbf{s}) \leq V^{k+1}(\mathbf{s}) \ \forall \mathbf{s} \in \mathcal{D}$ if*

$$\mathbb{E}_{\pi_{\hat{Q}^k}(\mathbf{a}|\mathbf{s})} \left[ \frac{\pi_{\hat{Q}^k}(\mathbf{a}|\mathbf{s})}{\hat{\pi}_\beta(\mathbf{a}|\mathbf{s})} - 1 \right] \geq \max_{\mathbf{a} \ s.t. \ \hat{\pi}_\beta(\mathbf{a}|\mathbf{s}) > 0} \left( \frac{\pi_{\hat{Q}^k}(\mathbf{a}|\mathbf{s})}{\hat{\pi}_\beta(\mathbf{a}|\mathbf{s})} \right) \cdot \varepsilon.$$

The LHS of this inequality is equal to the amount of conservatism induced in the value, $\hat{V}^{k+1}$ in iteration $k+1$ of the CQL update, if the learned policy were equal to soft-optimal policy for $\hat{Q}^k$, i.e., when $\hat{\pi}^{k+1} = \pi_{\hat{Q}^k}$. However, as the actual policy, $\hat{\pi}^{k+1}$, may be different, the RHS is the maximal amount of potential overestimation due to this difference. To get a lower bound, we require the amount of underestimation to be higher, which is obtained if $\varepsilon$ is small, i.e. the policy changes slowly.

Our final result shows that CQL Q-function update is "gap-expanding", by which we mean that the difference in Q-values at in-distribution actions and over-optimistically erroneous out-of-distribution actions is higher than the corresponding difference under the actual Q-function. This implies that the policy $\pi^k(\mathbf{a}|\mathbf{s}) \propto \exp(\hat{Q}^k(\mathbf{s}, \mathbf{a}))$, is constrained to be closer to the dataset distribution, $\hat{\pi}_\beta(\mathbf{a}|\mathbf{s})$, thus the CQL update implicitly prevents the detrimental effects of OOD action and distribution shift, which has been a major concern in offline RL settings [30, 34, 15].

**Theorem 3.4** (CQL is gap-expanding). *At any iteration $k$, CQL expands the difference in expected Q-values under the behavior policy $\pi_\beta(\mathbf{a}|\mathbf{s})$ and $\mu_k$, such that for large enough values of $\alpha_k$, we have that $\forall \mathbf{s} \in \mathcal{D}$, $\mathbb{E}_{\pi_\beta(\mathbf{a}|\mathbf{s})}[\hat{Q}^k(\mathbf{s}, \mathbf{a})] - \mathbb{E}_{\mu_k(\mathbf{a}|\mathbf{s})}[\hat{Q}^k(\mathbf{s}, \mathbf{a})] > \mathbb{E}_{\pi_\beta(\mathbf{a}|\mathbf{s})}[Q^k(\mathbf{s}, \mathbf{a})] - \mathbb{E}_{\mu_k(\mathbf{a}|\mathbf{s})}[Q^k(\mathbf{s}, \mathbf{a})]$.*

When function approximation or sampling error makes OOD actions have higher learned Q-values, CQL backups are expected to be more robust, in that the policy is updated using Q-values that prefer in-distribution actions. As we will empirically show in Appendix B, prior offline RL methods that do not explicitly constrain or regularize the Q-function may not enjoy such robustness properties.

**To summarize**, we showed that the CQL RL algorithm learns lower-bound Q-values with large enough $\alpha$, meaning that the final policy attains *at least* the estimated value. We also showed that the Q-function is *gap-expanding*, meaning that it should only ever *over-estimate* the gap between in-distribution and out-of-distribution actions, preventing OOD actions.

### 3.3 Safe Policy Improvement Guarantees

In Section 3.1 we proposed novel objectives for Q-function training such that the expected value of a policy under the resulting Q-function lower bounds the actual performance of the policy. In Section 3.2, we used the learned conservative Q-function for policy improvement. In this section, we show that this procedure actually optimizes a well-defined objective and provide a safe policy improvement result for CQL, along the lines of Theorems 1 and 2 in Laroche et al. [33].

To begin with, we define *empirical return* of any policy $\pi$, $J(\pi, \hat{M})$, which is equal to the discounted return of a policy $\pi$ in the *empirical* MDP, $\hat{M}$, that is induced by the transitions observed in the dataset $\mathcal{D}$, i.e. $\hat{M} = \{s, a, r, s' \in \mathcal{D}\}$. $J(\pi, M)$ refers to the expected discounted return attained by a policy $\pi$ in the actual underlying MDP, $M$. In Theorem 3.5, we first show that CQL (Equation 2) optimizes a well-defined penalized RL empirical objective. All proofs are found in Appendix D.4.

**Theorem 3.5.** *Let $\hat{Q}^\pi$ be the fixed point of Equation 2, then $\pi^*(\mathbf{a}|\mathbf{s}) := \arg\max_\pi \mathbb{E}_{\mathbf{s}\sim\rho(\mathbf{s})}[\hat{V}^\pi(\mathbf{s})]$ is equivalently represented as: $\pi^*(\mathbf{a}|\mathbf{s}) \leftarrow \arg\max_\pi \ J(\pi, \hat{M}) - \alpha\frac{1}{1-\gamma}\mathbb{E}_{\mathbf{s}\sim d_{\hat{M}}^\pi(\mathbf{s})}[D_{CQL}(\pi, \hat{\pi}_\beta)(\mathbf{s})],$ where $D_{CQL}(\pi, \pi_\beta)(\mathbf{s}) := \sum_{\mathbf{a}} \pi(\mathbf{a}|\mathbf{s}) \cdot \left(\frac{\pi(\mathbf{a}|\mathbf{s})}{\pi_\beta(\mathbf{a}|\mathbf{s})} - 1\right).$*

Intuitively, Theorem 3.5 says that CQL optimizes the return of a policy in the empirical MDP, $\hat{M}$, while also ensuring that the learned policy $\pi$ is not too different from the behavior policy, $\hat{\pi}_\beta$ via a penalty that depends on $D_{\text{CQL}}$. Note that this penalty is implicitly introduced by virtue by the gap-expanding (Theorem 3.4) behavior of CQL. Next, building upon Theorem 3.5 and the analysis of CPO [1], we show that CQL provides a $\zeta$-safe policy improvement over $\hat{\pi}_\beta$.

**Theorem 3.6.** *Let $\pi^*(\mathbf{a}|\mathbf{s})$ be the policy obtained in Theorem 3.5. Then, the policy $\pi^*(\mathbf{a}|\mathbf{s})$ is a $\zeta$-safe policy improvement over $\hat{\pi}_\beta$ in the actual MDP $M$, i.e., $J(\pi^*, M) \geq J(\hat{\pi}_\beta, M) - \zeta$ with high probability $1 - \delta$, where $\zeta$ is given by,*

$$\zeta = \mathcal{O}\left(\frac{\gamma}{(1-\gamma)^2}\right) \mathbb{E}_{\mathbf{s}\sim d_{\hat{M}}^{\pi^*}(\mathbf{s})}\left[\frac{\sqrt{|\mathcal{A}|}}{\sqrt{|\mathcal{D}(\mathbf{s})|}}\sqrt{D_{CQL}(\pi^*, \hat{\pi}_\beta)(\mathbf{s})+1}\right] - \underbrace{\left(J(\pi^*, \hat{M}) - J(\hat{\pi}_\beta, \hat{M})\right)}_{\geq \frac{\alpha}{1-\gamma}\mathbb{E}_{\mathbf{s}\sim d_{\hat{M}}^{\pi^*}(\mathbf{s})}[D_{CQL}(\pi^*, \hat{\pi}_\beta)(\mathbf{s})]}$$

The expression of $\zeta$ in Theorem 3.6 consists of two terms: the first term captures the decrease in policy performance in $M$, that occurs due to the mismatch between $\hat{M}$ and $M$, also referred to as *sampling error*. The second term captures the increase in policy performance due to CQL in empirical MDP, $\hat{M}$. The policy $\pi^*$ obtained by optimizing $\pi$ against the CQL Q-function improves upon the behavior policy, $\hat{\pi}_\beta$ for suitably chosen values of $\alpha$. When sampling error is small, i.e., $|\mathcal{D}(\mathbf{s})|$ is large, then smaller values of $\alpha$ are enough to provide an improvement over the behavior policy.

**To summarize,** CQL optimizes a well-defined, penalized empirical RL objective, and performs high-confidence safe policy improvement over the behavior policy. The extent of improvement is negatively influenced by higher sampling error, which decays as more samples are observed.

## 4 Practical Algorithm and Implementation Details

We now describe two practical offline deep reinforcement learning methods based on CQL: an actor-critic variant and a Q-learning variant. Pseudocode is shown in Algorithm 1, with differences from conventional actor-critic algorithms (e.g., SAC [19]) and deep Q-learning algorithms (e.g., DQN [39]) in red. Our algorithm uses the CQL($\mathcal{H}$) (or CQL($\mathcal{R}$) in general) objective from the CQL framework

---

**Algorithm 1** Conservative Q-Learning (both variants)

1: Initialize Q-function, $Q_\theta$, and optionally a policy, $\pi_\phi$.
2: **for** step $t$ in $\{1, \ldots, N\}$ **do**
3:     Train the Q-function using $G_Q$ gradient steps on objective from Equation 4
    $\theta_t := \theta_{t-1} - \eta_Q \nabla_\theta \text{CQL}(\mathcal{R})(\theta)$
    (Use $\mathcal{B}^*$ for Q-learning, $\mathcal{B}^{\pi_{\phi_t}}$ for actor-critic)
4:     (only with actor-critic) Improve policy $\pi_\phi$ via $G_\pi$ gradient steps on $\phi$ with SAC-style entropy regularization:
    $\phi_t := \phi_{t-1} + \eta_\pi \mathbb{E}_{\mathbf{s}\sim\mathcal{D}, \mathbf{a}\sim\pi_\phi(\cdot|\mathbf{s})}[Q_\theta(\mathbf{s}, \mathbf{a}) - \log\pi_\phi(\mathbf{a}|\mathbf{s})]$
5: **end for**

---

for training the Q-function $Q_\theta$, which is parameterized by a neural network with parameters $\theta$. For the actor-critic algorithm, a policy $\pi_\phi$ is trained as well. Our algorithm modifies the objective for the Q-function (swaps out Bellman error with CQL($\mathcal{H}$)) or CQL($\rho$) in a standard actor-critic or Q-learning setting, as shown in Line 3. As discussed in Section 3.2, due to the explicit penalty on the Q-function, CQL methods do not use a policy constraint, unlike prior offline RL methods [30, 59, 54, 34]. Hence, we do not require fitting an additional behavior policy estimator, simplifying our method.

**Implementation details.** Our algorithm requires an addition of only **20** lines of code on top of standard implementations of soft actor-critic (SAC) [19] for continuous control experiments and on top of QR-DQN [8] for the discrete control. The tradeoff factor, $\alpha$ is fixed at constant values described in Appendix F for gym tasks and discrete control and is automatically tuned via Lagrangian dual gradient descent for other domains. We use default hyperparameters from SAC, except that the learning rate for the policy was chosen from {3e-5, 1e-4, 3e-4}, and is less than or equal to the Q-function, as dictated by Theorem 3.3. Elaborate details are provided in Appendix F.

## 5 Related Work

We now briefly discuss prior work in offline RL and off-policy evaluation, comparing and contrasting these works with our approach. More technical discussion of related work is provided in Appendix E.

**Off-policy evaluation (OPE).** Several different paradigms have been used to perform off-policy evaluation. Earlier works [51, 49, 52] used per-action importance sampling on Monte-Carlo returns to obtain an OPE return estimator. Recent approaches [36, 17, 40, 60] use marginalized importance sampling by directly estimating the state-distribution importance ratios via some form of dynamic programming [34] and typically exhibit less variance than per-action importance sampling at the cost of bias. Because these methods use dynamic programming, they can suffer from OOD actions [34, 17, 20, 40]. In contrast, the regularizer in CQL explicitly addresses the impact of OOD actions due to its gap-expanding behavior, and obtains conservative value estimates.

**Offline RL.** As discussed in Section 2, offline Q-learning methods suffer from issues pertaining to OOD actions. Prior works have attempted to solve this problem by constraining the learned policy to be "close" to the behavior policy, for example as measured by KL-divergence [27, 59, 48, 54], Wasserstein distance [59], or MMD [30], and then only using actions sampled from this constrained policy in the Bellman backup or applying a value penalty. SPIBB [33, 41] methods bootstrap using the behavior policy in a Q-learning algorithm for unseen actions. Most of these methods require a separately estimated model to the behavior policy, $\pi_\beta(\mathbf{a}|\mathbf{s})$ [15, 30, 59, 27, 54, 55], and are thus limited by their ability to accurately estimate the unknown behavior policy [42], which might be especially complex in settings where the data is collected from multiple sources [34]. In contrast, CQL does not require estimating the behavior policy. Prior work has explored some forms of Q-function penalties [23, 58], but only in the standard online RL setting with demonstrations. Luo et al. [38] learn a conservatively-extrapolated value function by enforcing a linear extrapolation property over the state-space, and a learned dynamics model to obtain policies for goal-reaching tasks. Kakade and Langford [28] proposed the CPI algorithm, that improves a policy conservatively in online RL.

Alternate prior approaches to offline RL estimate some sort of uncertainty to determine the trustworthiness of a Q-value prediction [30, 3, 34], typically using uncertainty estimation techniques from exploration in online RL [47, 26, 46, 7]. These methods have not been generally performant in offline RL [15, 30, 34] due to the high-fidelity requirements of uncertainty estimates in offline RL [34]. Robust MDPs [24, 50, 56, 44] have been a popular theoretical abstraction for offline RL, but tend to be highly conservative in policy improvement. We expect CQL to be less conservative since CQL does not underestimate Q-values for all state-action tuples. Works on high confidence policy improvement [57] provides safety guarantees for improvement but tend to be conservative. The gap-expanding property of CQL backups, shown in Theorem 3.4, is related to how gap-increasing Bellman backup operators [6, 37] are more robust to estimation error in online RL.

**Theoretical results.** Our theoretical results (Theorems 3.5, 3.6) are related to prior work on safe policy improvement [33, 50], and a direct comparison to Theorems 1 and 2 in Laroche et al. [33] suggests similar quadratic dependence on the horizon and an inverse square-root dependence on the counts. Our bounds improve over the $\infty$-norm bounds in Petrik et al. [50].

# 6 Experimental Evaluation

We compare CQL to prior offline RL methods on a range of domains and dataset compositions, including continuous and discrete action spaces, state observations of varying dimensionality, and high-dimensional image inputs. We first evaluate actor-critic CQL, using CQL($\mathcal{H}$) from Algorithm 1, on continuous control datasets from the D4RL benchmark [12]. We compare to: prior offline RL methods that use a policy constraint – BEAR [30] and BRAC [59]; SAC [19], an off-policy actor-critic method that we adapt to offline setting; and behavioral cloning (BC).

**Gym domains.** Results for the gym domains are shown in Table 1. The results for BEAR, BRAC, SAC, and BC are based on numbers reported by Fu et al. [12]. On the datasets generated from a single policy, marked as "-random", "-expert" and "-medium", CQL roughly matches or exceeds the best prior methods, but by a small margin. However, on datasets that combine multiple policies ("-mixed", "-medium-expert" and "-random-expert"), that are more likely to be common in practical datasets, CQL outperforms prior methods by large margins, sometimes as much as **2-3x**.

**Adroit tasks.** The more complex Adroit [53] tasks (shown on the right) in D4RL require controlling a 24-DoF robotic hand, using limited data from human demonstrations. These tasks are substantially more difficult than the gym tasks in terms of both the dataset composition and high dimensionality. Prior offline RL methods generally struggle to learn meaningful behaviors on these tasks, and the strongest baseline is BC. As shown in Table 2, CQL variants are the only 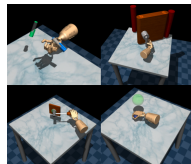 methods that improve over BC, attaining scores that are **2-9x** those of the next best offline RL method.

| Task Name | SAC | BC | BEAR | BRAC-p | BRAC-v | CQL($\mathcal{H}$) |
|---|---|---|---|---|---|---|
| halfcheetah-random | 30.5 | 2.1 | 25.5 | 23.5 | 28.1 | **35.4** |
| hopper-random | **11.3** | 9.8 | 9.5 | **11.1** | **12.0** | 10.8 |
| walker2d-random | 4.1 | 1.6 | **6.7** | 0.8 | 0.5 | **7.0** |
| halfcheetah-medium | -4.3 | 36.1 | 38.6 | **44.0** | **45.5** | **44.4** |
| walker2d-medium | 0.9 | 6.6 | 33.2 | 72.7 | **81.3** | 74.5 |
| hopper-medium | 0.8 | 29.0 | 47.6 | 31.2 | 32.3 | **86.6** |
| halfcheetah-expert | -1.9 | **107.0** | **108.2** | 3.8 | -1.1 | 104.8 |
| hopper-expert | 0.7 | **109.0** | **110.3** | 6.6 | 3.7 | **109.9** |
| walker2d-expert | -0.3 | **125.7** | 106.1 | -0.2 | -0.0 | 121.6 |
| halfcheetah-medium-expert | 1.8 | 35.8 | 51.7 | 43.8 | 45.3 | **62.4** |
| walker2d-medium-expert | 1.9 | 11.3 | 10.8 | -0.3 | 0.9 | **98.7** |
| hopper-medium-expert | 1.6 | **111.9** | 4.0 | 1.1 | 0.8 | **111.0** |
| halfcheetah-random-expert | 53.0 | 1.3 | 24.6 | 30.2 | 2.2 | **92.5** |
| walker2d-random-expert | 0.8 | 0.7 | 1.9 | 0.2 | 2.7 | **91.1** |
| hopper-random-expert | 5.6 | 10.1 | 10.1 | 5.8 | 11.1 | **110.5** |
| halfcheetah-medium-replay | -2.4 | 38.4 | 36.2 | **45.6** | **45.9** | 46.2 |
| hopper-medium-replay | 3.5 | 11.8 | 25.3 | 0.7 | 0.8 | **48.6** |
| walker2d-medium-replay | 1.9 | 11.3 | 10.8 | -0.3 | 0.9 | **32.6** |

Table 1: Performance of CQL($\mathcal{H}$) and prior methods on gym domains from D4RL, on the normalized return metric, averaged over 4 seeds. Note that CQL performs similarly or better than the best prior method with simple datasets, and greatly outperforms prior methods with complex distributions ("–mixed", "–random-expert", "–medium-expert").

CQL($\rho$) with $\rho = \hat{\pi}^{k-1}$ (the previous policy) outperforms CQL($\mathcal{H}$) on a number of these tasks, due to the higher action dimensionality resulting in higher variance for the CQL($\mathcal{H}$) importance weights. Both variants outperform prior methods.

| Domain | Task Name | BC | SAC | BEAR | BRAC-p | BRAC-v | CQL($\mathcal{H}$) | CQL($\rho$) |
|---|---|---|---|---|---|---|---|---|
| AntMaze | antmaze-umaze | 65.0 | 0.0 | **73.0** | 50.0 | 70.0 | **74.0** | **73.5** |
| | antmaze-umaze-diverse | 55.0 | 0.0 | 61.0 | 40.0 | 70.0 | **84.0** | 61.0 |
| | antmaze-medium-play | 0.0 | 0.0 | 0.0 | 0.0 | 0.0 | **61.2** | 4.6 |
| | antmaze-medium-diverse | 0.0 | 0.0 | 8.0 | 0.0 | 0.0 | **53.7** | 5.1 |
| | antmaze-large-play | 0.0 | 0.0 | 0.0 | 0.0 | 0.0 | **15.8** | 3.2 |
| | antmaze-large-diverse | 0.0 | 0.0 | 0.0 | 0.0 | 0.0 | **14.9** | 2.3 |
| Adroit | pen-human | 34.4 | 6.3 | -1.0 | 8.1 | 0.6 | 37.5 | **55.8** |
| | hammer-human | 1.5 | 0.5 | 0.3 | 0.3 | 0.2 | **4.4** | 2.1 |
| | door-human | 0.5 | 3.9 | -0.3 | -0.3 | -0.3 | **9.9** | **9.1** |
| | relocate-human | 0.0 | 0.0 | -0.3 | -0.3 | -0.3 | 0.20 | **0.35** |
| | pen-cloned | **56.9** | 23.5 | 26.5 | 1.6 | -2.5 | 39.2 | 40.3 |
| | hammer-cloned | 0.8 | 0.2 | 0.3 | 0.3 | 0.3 | 2.1 | **5.7** |
| | door-cloned | -0.1 | 0.0 | -0.1 | -0.1 | -0.1 | 0.4 | **3.5** |
| | relocate-cloned | **-0.1** | -0.2 | -0.3 | -0.3 | -0.3 | **-0.1** | **-0.1** |
| Kitchen[0] | kitchen-complete | 33.8 | 15.0 | 0.0 | 0.0 | 0.0 | **43.8** | 31.3 |
| | kitchen-partial | 33.8 | 0.0 | 13.1 | 0.0 | 0.0 | **49.8** | **50.1** |
| | kitchen-undirected | 47.5 | 2.5 | 47.2 | 0.0 | 0.0 | **51.0** | **52.4** |

Table 2: Normalized scores of all methods on AntMaze, Adroit, and kitchen domains from D4RL, averaged across 4 seeds. On the harder mazes, CQL is the *only* method that attains non-zero returns, and is the only method to outperform simple behavioral cloning on Adroit tasks with human demonstrations. We observed that the CQL($\rho$) variant, which avoids importance weights, trains more stably, with no sudden fluctuations in policy performance over the course of training, on the higher-dimensional Adroit tasks.

**AntMaze.** These D4RL tasks require composing parts of suboptimal trajectories to form more optimal policies for reaching goals on a MuJoco Ant robot. Prior methods make some progress on the simpler U-maze, but only CQL is able to make meaningful progress on the much harder medium and large mazes, outperforming prior methods by a very wide margin.

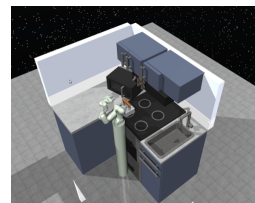

**Kitchen tasks.** Next, we evaluate CQL on the Franka kitchen domain [18] from D4RL [14]. The goal is to control a 9-DoF robot to manipulate multiple objects (microwave, kettle, etc.) *sequentially*, in a single episode to reach a desired configuration, with only sparse 0-1 completion reward for every object that attains the target configuration. These tasks are especially challenging, since they require composing parts of trajectories, precise long-horizon manipulation, and handling human-provided teleoperation data. As shown in Table 2, CQL outperforms prior methods in this setting, and is the only method that outperforms behavioral cloning, attaining over **40%** success rate on all tasks.

**Offline RL on Atari games.** Lastly, we evaluate a discrete-action Q-learning variant of CQL (Algorithm 1) on offline, image-based Atari games [5]. We compare CQL to REM [3] and QR-DQN [8] on the five Atari tasks (Pong, Breakout, Qbert, Seaquest and Asterix) that are evaluated in detail by Agarwal et al. [3], using the dataset released by the authors.

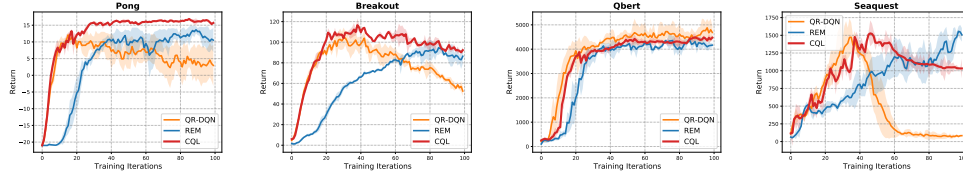

Figure 1: Performance of CQL, QR-DQN and REM as a function of training steps (x-axis) in setting (1) when provided with only the first 20% of the samples of an online DQN run. Note that CQL is able to learn stably on 3 out of 4 games, and its performance does not degrade as steeply as QR-DQN on Seaquest.

Following the evaluation protocol of Agarwal et al. [3], we evaluated on two types of datasets, both of which were generated from the DQN-replay dataset, released by [3]: (1) a dataset consisting of the first 20% of the samples observed by an online DQN agent and (2) datasets consisting of only 1% and 10% of all samples observed by an online DQN agent (Figures 6 and 7 in [3]). In setting (1), shown in Figure 1, CQL generally achieves similar or better performance throughout as QR-DQN and REM. When only using only 1% or 10% of the data, in setting (2) (Table 3), CQL substantially outperforms REM and QR-DQN, especially in the harder 1% condition, achieving **36x** and **6x** times the return of the best prior method on Q*bert and Breakout, respectively.

**Analysis of CQL.** Finally, we perform empirical evaluation to verify that CQL indeed lower-bounds the value function, thus verifying Theorems 3.2, Appendix D.1 empirically. To this end, we estimate the average value of the learned policy predicted by CQL, $\mathbb{E}_{\mathbf{s}\sim\mathcal{D}}[\hat{V}^k(\mathbf{s})]$, and report the difference against the actual discounted return of the policy $\pi^k$ in Table 4. We also estimate these values for baselines, including the minimum predicted Q-value under an ensemble [19, 16] of Q-functions with varying ensemble sizes, which is a stan-

| Task Name | QR-DQN | REM | CQL($\mathcal{H}$) |
|---|---|---|---|
| Pong (1%) | -13.8 | -6.9 | **19.3** |
| Breakout | 7.9 | 11.0 | **61.1** |
| Q*bert | 383.6 | 343.4 | **14012.0** |
| Seaquest | 672.9 | 499.8 | **779.4** |
| Asterix | 166.3 | 386.5 | **592.4** |
| Pong (10%) | 15.1 | 8.9 | **18.5** |
| Breakout | 151.2 | 86.7 | **269.3** |
| Q*bert | 7091.3 | 8624.3 | **13855.6** |
| Seaquest | 2984.8 | **3936.6** | 3674.1 |
| Asterix | **189.2** | 75.1 | 156.3 |

Table 3: CQL, REM and QR-DQN in setting (1) with 1% data (top), and 10% data (bottom). CQL drastically outperforms prior methods with 1% data, and usually attains better performance with 10% data.

dard technique to prevent overestimed Q-values [16, 19, 21] and BEAR [30], a policy constraint method. The results show that CQL learns a lower bound for all three tasks, whereas the baselines are prone to overestimation. We also evaluate a variant of CQL that uses Equation 1, and observe that the resulting values are lower (that is, underestimate the true values) as compared to CQL($\mathcal{H}$). This provides empirical evidence that CQL($\mathcal{H}$) attains a tighter lower bound than the point-wise bound in Equation 1, as per Theorem 3.2. We also present an empirical analysis to show that Theorem 3.4, that CQL is gap-expanding, holds in practice in Appendix B, and present an ablation study on various design choices used in CQL in Appendix G.

| Task Name | CQL($\mathcal{H}$) | CQL (Eqn. 1) | Ensemble(2) | Ens.(4) | Ens.(10) | Ens.(20) | BEAR |
|---|---|---|---|---|---|---|---|
| hopper-medium-expert | **-43.20** | -151.36 | 3.71e6 | 2.93e6 | 0.32e6 | 24.05e3 | 65.93 |
| hopper-mixed | **-10.93** | -22.87 | 15.00e6 | 59.93e3 | 8.92e3 | 2.47e3 | 1399.46 |
| hopper-medium | **-7.48** | -156.70 | 26.03e12 | 437.57e6 | 1.12e12 | 885e3 | 4.32 |

Table 4: Difference between predicted policy values and the true policy value for CQL, a variant of CQL that uses Equation 1, the minimum of an ensemble of varying sizes, and BEAR [30] on three D4RL datasets. CQL is the only method that lower-bounds the actual return (i.e., has negative differences), and CQL($\mathcal{H}$) is much less conservative than CQL (Eqn. 1).

## 7   Discussion

We proposed conservative Q-learning (CQL), an algorithmic framework for offline RL that learns a lower bound on the policy value. Empirically, we demonstrate that CQL outperforms prior offline RL methods on a wide range of offline RL benchmark tasks, including complex control tasks and tasks with raw image observations. In many cases, the performance of CQL is substantially better than the best-performing prior methods, exceeding their final returns by 2-5x. The simplicity and efficacy of CQL make it a promising choice for a wide range of real-world offline RL problems. However, a number of challenges remain. While we prove that CQL learns lower bounds on the Q-function in the tabular, linear, and a subset of non-linear function approximation cases, a rigorous theoretical analysis of CQL with deep neural nets, is left for future work. Additionally, offline RL methods are liable to suffer from overfitting in the same way as standard supervised methods, so another important challenge for future work is to devise simple and effective early stopping methods, analogous to validation error in supervised learning.

## Broader Impact

Offline RL offers the promise to scale autonomous learning-based methods for decision-making to large-scale, real-world sequential decision making problems. Such methods can effectively leverage prior datasets without any further interaction, thus avoiding the exploration bottleneck and alleviating many of the safety and cost constraints associated with online reinforcement learning. In this work, we proposed conservative Q-learning (CQL), an algorithmic framework for offline reinforcement learning that learns a Q-function such that the expected policy value under this learned Q-function lower-bounds the actual policy value. This mitigates value function over-estimation issues due to distributional shift, which in practice are one of the major challenges in offline reinforcement learning. We analyzed algorithms derived from the CQL framework and demonstrated their performance empirically.

Our primary aim behind this work is to develop simple and effective offline RL algorithms, and we believe that CQL makes an important step in that direction. CQL can be applied directly to several problems of practical interest where large-scale datasets are abundant: autonomous driving, robotics, and software systems (such as recommender systems). We believe that a strong offline RL algorithm, coupled with highly expressive and powerful deep neural networks, will provide us the ability successfully apply end-to-end learning based approaches to such problems, providing considerable societal benefits. Of course, autonomous decision-making agents have a wide range of applications, and technology that enables more effective autonomous decision-making has both positive and negative societal effects. While effective autonomous decision-making can have considerable positive economic effects, it can also enable applications with complex implications in regard to privacy (e.g., in regard to autonomous agents on the web, recommendation agents, advertising, etc.), as well as complex economic effects due to changing economic conditions (e.g., changing job requirements, loss of jobs in some sectors and growth in others, etc.). Such implications apply broadly to technologies that enable automation and decision making, and are largely not unique to this specific work.

## Acknowledgements and Funding

We thank Mohammad Norouzi, Oleh Rybkin, Anton Raichuk, Avi Singh, Vitchyr Pong and anonymous reviewers from the Robotic AI and Learning Lab at UC Berkeley and NeurIPS for their feedback on an earlier version of this paper. We thank Rishabh Agarwal for help with the Atari QR-DQN/REM codebase and for sharing baseline results. This research was funded by the DARPA Assured Autonomy program, and compute support from Google and Amazon.

## Footnotes

[0]These results are with an older version of the Kitchen D4RL datasets [12], new results in the latest arXiv.

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
