[Supplementary Material]

# Appendices

## A    Discussion of CQL Variants

We derive several variants of CQL in Section 3.2. Here, we discuss these variants on more detail and describe their specific properties. We first derive the variants: CQL($\mathcal{H}$), CQL($\rho$), and then present another variant of CQL, which we call CQL(var). This third variant has strong connections to distributionally robust optimization [43].

**CQL($\mathcal{H}$).** In order to derive CQL($\mathcal{H}$), we substitute $\mathcal{R} = \mathcal{H}(\mu)$, and solve the optimization over $\mu$ in closed form for a given Q-function. For an optimization problem of the form:

$$\max_{\mu} \ \mathbb{E}_{\mathbf{x}\sim\mu(\mathbf{x})}[f(\mathbf{x})] + \mathcal{H}(\mu) \ \ \text{s.t.} \ \ \sum_{\mathbf{x}} \mu(\mathbf{x}) = 1, \ \mu(\mathbf{x}) \geq 0 \ \forall \mathbf{x},$$

the optimal solution is equal to $\mu^*(\mathbf{x}) = \frac{1}{Z}\exp(f(\mathbf{x}))$, where $Z$ is a normalizing factor. Plugging this into Equation 3, we exactly obtain Equation 4.

**CQL($\rho$).** In order to derive CQL($\rho$), we follow the above derivation, but our regularizer is a KL-divergence regularizer instead of entropy.

$$\max_{\mu} \ \mathbb{E}_{\mathbf{x}\sim\mu(\mathbf{x})}[f(\mathbf{x})] + D_{\mathrm{KL}}(\mu||\rho) \ \ \text{s.t.} \ \ \sum_{\mathbf{x}} \mu(\mathbf{x}) = 1, \ \mu(\mathbf{x}) \geq 0 \ \forall \mathbf{x}.$$

The optimal solution is given by, $\mu^*(\mathbf{x}) = \frac{1}{Z}\rho(\mathbf{x})\exp(f(\mathbf{x}))$, where $Z$ is a normalizing factor. Plugging this back into the CQL family (Equation 3), we obtain the following objective for training the Q-function (modulo some normalization terms):

$$\min_{Q} \ \alpha\mathbb{E}_{\mathbf{s}\sim d^{\pi_\beta}(\mathbf{s})}\left[\mathbb{E}_{\mathbf{a}\sim\rho(\mathbf{a}|\mathbf{s})}\left[Q(\mathbf{s},\mathbf{a})\frac{\exp(Q(\mathbf{s},\mathbf{a}))}{Z}\right] - \mathbb{E}_{\mathbf{a}\sim\pi_\beta(\mathbf{a}|\mathbf{s})}[Q(\mathbf{s},\mathbf{a})]\right] + \frac{1}{2}\mathbb{E}_{\mathbf{s},\mathbf{a},\mathbf{s}'\sim\mathcal{D}}\left[\left(Q - \mathcal{B}^{\pi_k}\hat{Q}^k\right)^2\right].$$

(5)

**CQL(var).** Finally, we derive a CQL variant that is inspired from the perspective of distributionally robust optimization (DRO) [43]. This version penalizes the variance in the Q-function across actions at all states $\mathbf{s}$, under some action-conditional distribution of our choice. In order to derive a canonical form of this variant, we invoke an identity from Namkoong and Duchi [43], which helps us simplify Equation 3. To start, we define the notion of "robust expectation": for any function $f(\mathbf{x})$, and any empirical distribution $\hat{P}(\mathbf{x})$ over a dataset $\{\mathbf{x}_1, \cdots, \mathbf{x}_N\}$ of $N$ elements, the "robust" expectation defined by:

$$R_N(\hat{P}) := \max_{\mu(\mathbf{x})} \ \mathbb{E}_{\mathbf{x}\sim\mu(\mathbf{x})}[f(\mathbf{x})] \ \ \text{s.t.} \ \ D_f(\mu(\mathbf{x}), \hat{P}(\mathbf{x})) \leq \frac{\delta}{N},$$

can be approximated using the following upper-bound:

$$R_N(\hat{P}) \leq \mathbb{E}_{\mathbf{x}\sim\hat{P}(\mathbf{x})}[f(\mathbf{x})] + \sqrt{\frac{2\delta \ \mathrm{var}_{\hat{P}(\mathbf{x})}(f(\mathbf{x}))}{N}},$$

(6)

where the gap between the two sides of the inequality decays inversely w.r.t. to the dataset size, $\mathcal{O}(1/N)$. By using Equation 6 to simplify Equation 3, we obtain an objective for training the Q-function that penalizes the variance of Q-function predictions under the distribution $\hat{P}$.

$$\min_{Q} \ \frac{1}{2}\mathbb{E}_{\mathbf{s},\mathbf{a},\mathbf{s}'\sim\mathcal{D}}\left[\left(Q - \mathcal{B}^{\pi_k}\hat{Q}^k\right)^2\right] + \alpha\mathbb{E}_{\mathbf{s}\sim d^{\pi_\beta}(\mathbf{s})}\left[\sqrt{\frac{\mathrm{var}_{\hat{P}(\mathbf{a}|\mathbf{s})}(Q(\mathbf{s},\mathbf{a}))}{d^{\pi_\beta}(s)|\mathcal{D}|}}\right]$$

$$+ \alpha\mathbb{E}_{s\sim d^{\pi_\beta}(\mathbf{s})}\left[\mathbb{E}_{\hat{P}(\mathbf{a}|\mathbf{s})}[Q(\mathbf{s},\mathbf{a})] - \mathbb{E}_{\pi_\beta(\mathbf{a}|\mathbf{s})}[Q(\mathbf{s},\mathbf{a})]\right] \quad (7)$$

The only remaining decision is the choice of $\hat{P}$, which can be chosen to be the inverse of the empirical action distribution in the dataset, $\hat{P}(\mathbf{a}|\mathbf{s}) \propto \frac{1}{\hat{D}(\mathbf{a}|\mathbf{s})}$, or even uniform over actions, $\hat{P}(\mathbf{a}|\mathbf{s}) = \mathrm{Unif}(\mathbf{a})$, to obtain this variant of variance-regularized CQL.

# B Discussion of Gap-Expanding Behavior of CQL Backups

In this section, we discuss in detail the consequences of the gap-expanding behavior of CQL backups over prior methods based on policy constraints that, as we show in this section, may not exhibit such gap-expanding behavior in practice. To recap, Theorem 3.4 shows that the CQL backup operator increases the difference between expected Q-value at in-distribution ($\mathbf{a} \sim \pi_\beta(\mathbf{a}|\mathbf{s})$) and out-of-distribution ($\mathbf{a}$ s.t. $\frac{\mu_k(\mathbf{a}|\mathbf{s})}{\pi_\beta(\mathbf{a}|\mathbf{s})} << 1$) actions. We refer to this property as the gap-expanding property of the CQL update operator.

**Function approximation may give rise to erroneous Q-values at OOD actions.** We start by discussing the behavior of prior methods based on policy constraints [30, 15, 27, 59] in the presence of function approximation. To recap, because computing the target value requires $\mathbb{E}_\pi[\hat{Q}(\mathbf{s}, \mathbf{a})]$, constraining $\pi$ to be close to $\pi_\beta$ will avoid evaluating $\hat{Q}$ on OOD actions. These methods typically do not impose any further form of regularization on the learned Q-function. Even with policy constraints, because function approximation used to represent the Q-function, learned Q-values at two distinct state-action pairs are coupled together. As prior work has argued and shown [2, 13, 31], the "generalization" or the coupling effects of the function approximator may be heavily influenced by the properties of the data distribution [13, 31]. For instance, Fu et al. [13] empirically shows that when the dataset distribution is narrow (i.e. state-action marginal entropy, $\mathcal{H}(d^{\pi_\beta}(\mathbf{s}, \mathbf{a}))$, is low [13]), the coupling effects of the Q-function approximator can give rise to incorrect Q-values at different states, though this behavior is absent without function approximation, and is not as severe with high-entropy (e.g. Uniform) state-action marginal distributions.

In offline RL, we will shortly present empirical evidence on high-dimensional MuJoCo tasks showing that certain dataset distributions, $\mathcal{D}$, may cause the learned Q-value at an OOD action $\mathbf{a}$ at a state $\mathbf{s}$, to in fact take on high values than Q-values at in-distribution actions at intermediate iterations of learning. This problem persists even when a large number of samples (e.g. $1M$) are provided for training, and the agent cannot correct these errors due to no active data collection.

Since actor-critic methods, including those with policy constraints, use the learned Q-function to train the policy, in an iterative online policy evaluation and policy improvement cycle, as discussed in Section 2, the errneous Q-function may push the policy towards OOD actions, especially when no policy constraints are used. Of course, policy constraints should prevent the policy from choosing OOD actions, however, as we will show that in certain cases, policy constraint methods might also fail to prevent the effects on the policy due to incorrectly high Q-values at OOD actions.

**How can CQL address this problem?** As we show in Theorem 3.4, the difference between expected Q-values at in-distribution actions and out-of-distribution actions is expanded by the CQL update. This property is a direct consequence of the specific nature of the CQL regularizer – that maximizes Q-values under the dataset distribution, and minimizes them otherwise. This difference depends upon the choice of $\alpha_k$, which can directly be controlled, since it is a free parameter. Thus, by effectively controlling $\alpha_k$, CQL can push down the learned Q-value at out-of-distribution actions as much is desired, correcting for the erroneous overestimation error in the process.

**Empirical evidence on high-dimensional benchmarks with neural networks.** We next empirically demonstrate the existence of of such Q-function estimation error on high-dimensional MuJoCo domains when deep neural network function approximators are used with stochastic optimization techniques. In order to measure this error, we plot the difference in expected Q-value under actions sampled from the behavior distribution, $\mathbf{a} \sim \pi_\beta(\mathbf{a}|\mathbf{s})$, and the maximum Q-value over actions sampled from a uniformly random policy, $\mathbf{a} \sim \text{Unif}(\mathbf{a}|\mathbf{s})$. That is, we plot the quantity

$$\hat{\Delta}^k = \mathbb{E}_{\mathbf{s},\mathbf{a}\sim\mathcal{D}} \left[ \max_{\mathbf{a}'_1, \cdots, \mathbf{a}'_N \sim \text{Unif}(\mathbf{a}')} [\hat{Q}^k(\mathbf{s}, \mathbf{a}')] - \hat{Q}^k(\mathbf{s}, \mathbf{a}) \right] \tag{8}$$

over the iterations of training, indexed by $k$. This quantity, intuitively, represents an estimate of the "advantage" of an action $\mathbf{a}$, under the Q-function, with respect to the optimal action $\max_{\mathbf{a}'} \hat{Q}^k(\mathbf{s}, \mathbf{a}')$. Since, we cannot perform exact maximization over the learned Q-function in a continuous action space to compute $\Delta$, we estimate it via sampling described in Equation 8.

We present these plots in Figure 2 on two datasets: hopper-expert and hopper-medium. The expert dataset is generated from a near-deterministic, expert policy, exhibits a narrow coverage of the state-action space, and limited to only a few directed trajectories. On this dataset, we find that $\hat{\Delta}^k$ is

(a) hopper-expert-v0                    (b) hopper-medium-v0

Figure 2: $\Delta^k$ as a function of training iterations for hopper-expert and hopper-medium datasets. Note that CQL (left) generally has negative values of $\Delta$, whereas BEAR (right) generally has positive $\Delta$ values, which also increase during training with increasing $k$ values.

always positive for the policy constraint method (Figure 2(a)) and increases during training – note, the continuous rise in $\hat{\Delta}^k$ values, in the case of the policy-constraint method, shown in Figure 2(a). This means that even if the dataset is generated from an expert policy, and policy constraints correct target values for OOD actions, incorrect Q-function generalization may make an out-of-distribution action appear promising. For the more stochastic hopper-medium dataset, that consists of a more diverse set of trajectories, shown in Figure 2(b), we still observe that $\hat{\Delta}^k > 0$ for the policy-constraint method, however, the relative magnitude is smaller than hopper-expert.

In contrast, Q-functions learned by CQL, generally satisfy $\hat{\Delta}^k < 0$, as is seen and these values are clearly smaller than those for the policy-constraint method. This provides some empirical evidence for Theorem 3.4, in that, the maximum Q-value at a randomly chosen action from the uniform distribution the action space is smaller than the Q-value at in-distribution actions.

On the hopper-expert task, as we show in Figure 2(a) (right), we eventually observe an "unlearning" effect, in the policy-constraint method where the policy performance deteriorates after a extra iterations in training. This "unlearning" effect is similar to what has been observed when standard off-policy Q-learning algorithms without any policy constraint are used in the offline regime [30, 34], on the other hand this effect is absent in the case of CQL, even after equally many training steps. The performance in the more-stochastic hopper-medium dataset fluctuates, but does not deteriorate suddenly.

To summarize this discussion, we concretely observed the following points via empirical evidence:

- CQL backups are gap expanding in practice, as justified by the negative $\hat{\Delta}^k$ values in Figure 2.

- Policy constraint methods, that do not impose any regularization on the Q-function may observe highly positive $\hat{\Delta}^k$ values during training, especially with narrow data distributions, indicating that gap-expansion may be absent.

- When $\hat{\Delta}^k$ values continuously grow during training, the policy might eventually suffer from an unlearning effect [34], as shown in Figure 2(a).

## C   Theorem Proofs

In this section, we provide proofs of the theorems in Sections 3.1 and 3.2. We first redefine notation for clarity and then provide the proofs of the results in the main paper.

**Notation.** Let $k \in \mathbb{N}$ denote an iteration of policy evaluation (in Section 3.1) or Q-iteration (in Section 3.2). In an iteration $k$, the objective – Equation 2 or Equation 3 – is optimized using the previous iterate (i.e. $\hat{Q}^{k-1}$) as the target value in the backup. $Q^k$ denotes the true, tabular Q-function iterate in the MDP, without any correction. In an iteration, say $k + 1$, the current tabular Q-function iterate, $Q^{k+1}$ is related to the previous tabular Q-function iterate $Q^k$ as: $Q^{k+1} = \mathcal{B}^\pi Q^k$ (for policy evaluation) or $Q^{k+1} = \mathcal{B}^{\pi_k} Q^k$ (for policy learning). Let $\hat{Q}^k$ denote the $k$-th Q-function iterate obtained from CQL. Let $\hat{V}^k$ denote the value function, $\hat{V}^k := \mathbb{E}_{\mathbf{a} \sim \pi(\mathbf{a}|\mathbf{s})}[\hat{Q}^k(\mathbf{s}, \mathbf{a})]$.

**A note on the value of** $\alpha$**.** Before proving the theorems, we remark that while the statements of Theorems 3.2, 3.1 and D.1 (we discuss this in Appendix D) show that CQL produces lower bounds if

$\alpha$ is larger than some threshold, so as to overcome either sampling error (Theorems 3.2 and 3.1) or function approximation error (Theorem D.1). While the optimal $\alpha_k$ in some of these cases depends on the current Q-value, $\hat{Q}^k$, we can always choose a worst-case value of $\alpha_k$ by using the inequality $\hat{Q}^k \leq 2R_{\max}/(1-\gamma)$, still guaranteeing a lower bound. If it is unclear why the learned Q-function $\hat{Q}^k$ should be bounded, we can always clamp the Q-values if they go outside $\left[\frac{-2R_{\max}}{1-\gamma}, \frac{2R_{\max}}{1-\gamma}\right]$.

We first prove Theorem 3.1, which shows that policy evaluation using a simplified version of CQL (Equation 1) results in a point-wise lower-bound on the Q-function.

**Proof of Theorem 3.1.** In order to start, we first note that the form of the resulting Q-function iterate, $\hat{Q}^k$, in the setting without function approximation. By setting the derivative of Equation 1 to 0, we obtain the following expression for $\hat{Q}^{k+1}$ in terms of $\hat{Q}^k$,

$$\forall \mathbf{s}, \mathbf{a} \in \mathcal{D}, k, \ \hat{Q}^{k+1}(\mathbf{s}, \mathbf{a}) = \hat{\mathcal{B}}^\pi \hat{Q}^k(\mathbf{s}, \mathbf{a}) - \alpha \frac{\mu(\mathbf{a}|\mathbf{s})}{\hat{\pi}_\beta(\mathbf{a}|\mathbf{s})}. \tag{9}$$

Now, since, $\mu(\mathbf{a}|\mathbf{s}) > 0, \alpha > 0, \hat{\pi}_\beta(\mathbf{a}|\mathbf{s}) > 0$, we observe that at each iteration we underestimate the next Q-value iterate, i.e. $\hat{Q}^{k+1} \leq \hat{\mathcal{B}}^\pi \hat{Q}^k$.

**Accounting for sampling error.** Note that so far we have only shown that the Q-values are upper-bounded by the the "empirical Bellman targets" given by, $\hat{\mathcal{B}}^\pi \hat{Q}^k$. In order to relate $\hat{Q}^k$ to the true Q-value iterate, $Q^k$, we need to relate the empirical Bellman operator, $\hat{\mathcal{B}}^\pi$ to the actual Bellman operator, $\mathcal{B}^\pi$. In Appendix D.3, we show that if the reward function $r(\mathbf{s}, \mathbf{a})$ and the transition function, $T(\mathbf{s}'|\mathbf{s}, \mathbf{a})$ satisfy "concentration" properties, meaning that the difference between the observed reward sample, $r (\mathbf{s}, \mathbf{a}, r, \mathbf{s}') \in \mathcal{D})$ and the actual reward function $r(\mathbf{s}, \mathbf{a})$ (and analogously for the transition matrix) is bounded with high probability, then overestimation due to the empirical Backup operator is bounded. Formally, with high probability (w.h.p.) $\geq 1 - \delta, \delta \in (0, 1)$,

$$\forall Q, \mathbf{s}, \mathbf{a} \in \mathcal{D}, \ \left|\hat{\mathcal{B}}^\pi Q(\mathbf{s}, \mathbf{a}) - \mathcal{B}^\pi Q(\mathbf{s}, \mathbf{a})\right| \leq \frac{C_{r,T,\delta} R_{\max}}{(1-\gamma)\sqrt{|\mathcal{D}(\mathbf{s}, \mathbf{a})|}}.$$

Hence, the following can be obtained, w.h.p.:

$$\hat{Q}^{k+1}(\mathbf{s}, \mathbf{a}) = \mathcal{B}^\pi \hat{Q}^k(\mathbf{s}, \mathbf{a}) \leq \mathcal{B}^\pi \hat{Q}^k(\mathbf{s}, \mathbf{a}) - \alpha \frac{\mu(\mathbf{a}|\mathbf{s})}{\hat{\pi}_\beta(\mathbf{a}|\mathbf{s})} + \frac{C_{r,T,\delta} R_{\max}}{(1-\gamma)\sqrt{|\mathcal{D}(\mathbf{s}, \mathbf{a})|}}. \tag{10}$$

Now we need to reason about the fixed point of the update procedure in Equation 9. The fixed point of Equation 9 is given by:

$$\hat{Q}^\pi \leq \mathcal{B}^\pi \hat{Q}^\pi - \alpha \frac{\mu(\mathbf{a}|\mathbf{s})}{\hat{\pi}_\beta(\mathbf{a}|\mathbf{s})} + \frac{C_{r,T,\delta} R_{\max}}{(1-\gamma)\sqrt{|\mathcal{D}(\mathbf{s}, \mathbf{a})|}} \implies \hat{Q}^\pi \leq (I - \gamma P^\pi)^{-1}\left[R - \alpha \frac{\mu}{\hat{\pi}_\beta} + \frac{C_{r,T,\delta} R_{\max}}{1-\gamma)\sqrt{|\mathcal{D}|}}\right]$$

$$\hat{Q}^\pi(\mathbf{s}, \mathbf{a}) \leq Q^\pi(\mathbf{s}, \mathbf{a}) - \alpha \left[(I - \gamma P^\pi)^{-1}\left[\frac{\mu}{\hat{\pi}_\beta}\right]\right](\mathbf{s}, \mathbf{a}) + \left[(I - \gamma P^\pi)^{-1}\frac{C_{r,T,\delta} R_{\max}}{(1-\gamma)\sqrt{|\mathcal{D}|}}\right](\mathbf{s}, \mathbf{a}),$$

thus proving the relationship in Theorem 3.1.

In order to guarantee a lower bound, $\alpha$ can be chosen to cancel any potential overestimation incurred by $\frac{C_{r,T,\delta} R_{\max}}{(1-\gamma)\sqrt{|\mathcal{D}|}}$. Note that this choice works, since $(I - \gamma P^\pi)^{-1}$ is a matrix with all non-negative entries. The choice of $\alpha$ that guarantees a lower bound is then given by:

$$\alpha \cdot \min_{\mathbf{s},\mathbf{a}}\left[\frac{\mu(\mathbf{a}|\mathbf{s})}{\hat{\pi}_\beta(\mathbf{a}|\mathbf{s})}\right] \geq \max_{\mathbf{s},\mathbf{a}} \frac{C_{r,T,\delta} R_{\max}}{(1-\gamma)\sqrt{|\mathcal{D}(\mathbf{s}, \mathbf{a})|}}$$

$$\implies \alpha \geq \max_{\mathbf{s},\mathbf{a}} \frac{C_{r,T,\delta} R_{\max}}{(1-\gamma)\sqrt{|\mathcal{D}(\mathbf{s}, \mathbf{a})|}} \cdot \max_{\mathbf{s},\mathbf{a}}\left[\frac{\mu(\mathbf{a}|\mathbf{s})}{\hat{\pi}_\beta(\mathbf{a}|\mathbf{s})}\right]^{-1}.$$

Note that the theoretically minimum possible value of $\alpha$ decreases as more samples are observed, i.e., when $|\mathcal{D}(\mathbf{s}, \mathbf{a})|$ is large. Also, note that since, $\frac{C_{r,T,\delta} R_{\max}}{(1-\gamma0\sqrt{|\mathcal{D}|}} \approx 0$, when $\hat{\mathcal{B}}^\pi = \mathcal{B}^\pi$, any $\alpha \geq 0$ guarantees a lower bound. And so choosing a value of $\alpha = 0$ is sufficient in this case.

Next, we prove Theorem 3.3 that shows that the additional term that maximizes the expected Q-value under the dataset distribution, $\mathbb{D}(\mathbf{s}, \mathbf{a})$, (or $d^{\pi_\beta}(\mathbf{s})\pi_\beta(\mathbf{a}|\mathbf{s})$, in the absence of sampling error), results in a lower-bound on only the expected value of the policy at a state, and not a pointwise lower-bound on Q-values at all actions.

**Proof of Theorem 3.2.** We first prove this theorem in the absence of sampling error, and then incorporate sampling error at the end, using a technique similar to the previous proof. In the tabular setting, we can set the derivative of the modified objective in Equation 2, and compute the Q-function update induced in the exact, tabular setting (this assumes $\hat{\mathcal{B}}^\pi = \mathcal{B}^\pi$) and $\pi_\beta(\mathbf{a}|\mathbf{s}) = \hat{\pi}_\beta(\mathbf{a}|\mathbf{s})$).

$$\forall \mathbf{s}, \mathbf{a}, k \ \hat{Q}^{k+1}(\mathbf{s}, \mathbf{a}) = \mathcal{B}^\pi \hat{Q}^k(\mathbf{s}, \mathbf{a}) - \alpha \left[ \frac{\mu(\mathbf{a}|\mathbf{s})}{\pi_\beta(\mathbf{a}|\mathbf{s})} - 1 \right]. \tag{11}$$

Note that for state-action pairs, $(\mathbf{s}, \mathbf{a})$, such that, $\mu(\mathbf{a}|\mathbf{s}) < \pi_\beta(\mathbf{a}|\mathbf{s})$, we are infact adding a positive quantity, $1 - \frac{\mu(\mathbf{a}|\mathbf{s})}{\pi_\beta(\mathbf{a}|\mathbf{s})}$, to the Q-function obtained, and this we cannot guarantee a point-wise lower bound, i.e. $\exists \mathbf{s}, \mathbf{a}$, s.t. $\hat{Q}^{k+1}(\mathbf{s}, \mathbf{a}) \geq Q^{k+1}(\mathbf{s}, \mathbf{a})$. To formally prove this, we can construct a counter-example three-state, two-action MDP, and choose a specific behavior policy $\pi(\mathbf{a}|\mathbf{s})$, such that this is indeed the case.

The value of the policy, on the other hand, $\hat{V}^{k+1}$ is underestimated, since:

$$\hat{V}^{k+1}(\mathbf{s}) := \mathbb{E}_{\mathbf{a}\sim\pi(\mathbf{a}|\mathbf{s})}\left[ \hat{Q}^{k+1}(\mathbf{s}, \mathbf{a}) \right] = \mathcal{B}^\pi \hat{V}^k(\mathbf{s}) - \alpha \mathbb{E}_{\mathbf{a}\sim\pi(\mathbf{a}|\mathbf{s})}\left[ \frac{\mu(\mathbf{a}|\mathbf{s})}{\pi_\beta(\mathbf{a}|\mathbf{s})} - 1 \right]. \tag{12}$$

and we can show that $D_{\text{CQL}}(\mathbf{s}) := \sum_{\mathbf{a}} \pi(\mathbf{a}|\mathbf{s}) \left[ \frac{\mu(\mathbf{a}|\mathbf{s})}{\pi_\beta(\mathbf{a}|\mathbf{s})} - 1 \right]$ is always positive, when $\pi(\mathbf{a}|\mathbf{s}) = \mu(\mathbf{a}|\mathbf{s})$. To note this, we present the following derivation:

$$\begin{aligned}
D_{\text{CQL}}(\mathbf{s}) :&= \sum_{\mathbf{a}} \pi(\mathbf{a}|\mathbf{s}) \left[ \frac{\mu(\mathbf{a}|\mathbf{s})}{\pi_\beta(\mathbf{a}|\mathbf{s})} - 1 \right] \\
&= \sum_{\mathbf{a}} (\pi(\mathbf{a}|\mathbf{s}) - \pi_\beta(\mathbf{a}|\mathbf{s}) + \pi_\beta(\mathbf{a}|\mathbf{s})) \left[ \frac{\mu(\mathbf{a}|\mathbf{s})}{\pi_\beta(\mathbf{a}|\mathbf{s})} - 1 \right] \\
&= \sum_{\mathbf{a}} (\pi(\mathbf{a}|\mathbf{s}) - \pi_\beta(\mathbf{a}|\mathbf{s})) \left[ \frac{\pi(\mathbf{a}|\mathbf{s}) - \pi_\beta(\mathbf{a}|\mathbf{s})}{\pi_\beta(\mathbf{a}|\mathbf{s}} \right] + \sum_{\mathbf{a}} \pi_\beta(\mathbf{a}|\mathbf{s}) \left[ \frac{\mu(\mathbf{a}|\mathbf{s})}{\pi_\beta(\mathbf{a}|\mathbf{s})} - 1 \right] \\
&= \sum_{\mathbf{a}} \underbrace{\left[ \frac{(\pi(\mathbf{a}|\mathbf{s}) - \pi_\beta(\mathbf{a}|\mathbf{s}))^2}{\pi_\beta(\mathbf{a}|\mathbf{s})} \right]}_{\geq 0} + \ 0 \ \ \text{since,} \ \sum_{\mathbf{a}} \pi(\mathbf{a}|\mathbf{s}) = \sum_{\mathbf{a}} \pi_\beta(\mathbf{a}|\mathbf{s}) = 1.
\end{aligned}$$

Note that the marked term, is positive since both the numerator and denominator are positive, and this implies that $D_{\text{CQL}}(\mathbf{s}) \geq 0$. Also, note that $D_{\text{CQL}}(\mathbf{s}) = 0$, iff $\pi(\mathbf{a}|\mathbf{s}) = \pi_\beta(\mathbf{a}|\mathbf{s})$. This implies that each value iterate incurs some underestimation, $\hat{V}^{k+1}(\mathbf{s}) \leq \mathcal{B}^\pi \hat{V}^k(\mathbf{s})$.

Now, we can compute the fixed point of the recursion in Equation 12, and this gives us the following estimated policy value:

$$\hat{V}^\pi(\mathbf{s}) = V^\pi(\mathbf{s}) - \alpha \left[ \underbrace{(I - \gamma P^\pi)^{-1}}_{\text{non-negative entries}} \underbrace{\mathbb{E}_\pi \left[ \frac{\pi}{\pi_\beta} - 1 \right]}_{\geq 0} \right](\mathbf{s}),$$

thus showing that in the absence of sampling error, Theorem 3.2 gives a lower bound. It is straightforward to note that this expression is tighter than the expression for policy value in Proposition 3.2, since, we explicitly subtract 1 in the expression of Q-values (in the exact case) from the previous proof.

**Incorporating sampling error.** To extend this result to the setting with sampling error, similar to the previous result, the maximal overestimation at each iteration $k$, is bounded by $\frac{C_{r,T,\delta} R_{\max}}{1-\gamma}$. The resulting value-function satisfies (w.h.p.), $\forall \mathbf{s} \in \mathcal{D}$,

$$\hat{V}^\pi(\mathbf{s}) \leq V^\pi(\mathbf{s}) - \alpha \left[ (I - \gamma P^\pi)^{-1} \mathbb{E}_\pi \left[ \frac{\pi}{\hat{\pi}_\beta} - 1 \right] \right](\mathbf{s}) + \left[ (I - \gamma P^\pi)^{-1} \frac{C_{r,T,\delta} R_{\max}}{(1-\gamma)\sqrt{|\mathcal{D}|}} \right](\mathbf{s})$$

thus proving the theorem statement. In this case, the choice of $\alpha$, that prevents overestimation w.h.p. is given by:

$$\alpha \geq \max_{\mathbf{s},\mathbf{a}\in\mathcal{D}} \frac{C_{r,T}R_{\max}}{(1-\gamma)\sqrt{|\mathcal{D}(\mathbf{s},\mathbf{a})|}} \cdot \max_{\mathbf{s}\in\mathcal{D}} \left[\sum_{\mathbf{a}} \pi(\mathbf{a}|\mathbf{s})\left(\frac{\pi(\mathbf{a}|\mathbf{s})}{\hat{\pi}_\beta(\mathbf{a}|\mathbf{s}))}-1\right)\right]^{-1}.$$

Similar to Theorem 3.1, note that the theoretically acceptable value of $\alpha$ decays as the number of occurrences of a state action pair in the dataset increases. Next we provide a proof for Theorem 3.3.

**Proof of Theorem 3.3.** In order to prove this theorem, we compute the difference induced in the policy value, $\hat{V}^{k+1}$, derived from the Q-value iterate, $\hat{Q}^{k+1}$, with respect to the previous iterate $\mathcal{B}^\pi \hat{Q}^k$. If this difference is negative at each iteration, then the resulting Q-values are guaranteed to lower bound the true policy value.

$$
\begin{aligned}
\mathbb{E}_{\hat{\pi}^{k+1}(\mathbf{a}|\mathbf{s})}[\hat{Q}^{k+1}(\mathbf{s},\mathbf{a})] &= \mathbb{E}_{\hat{\pi}^{k+1}(\mathbf{a}|\mathbf{s})}\left[\mathcal{B}^\pi\hat{Q}^k(\mathbf{s},\mathbf{a})\right] - \mathbb{E}_{\hat{\pi}^{k+1}(\mathbf{a}|\mathbf{s})}\left[\frac{\pi_{\hat{Q}^k}(\mathbf{a}|\mathbf{s})}{\hat{\pi}_\beta(\mathbf{a}|\mathbf{s})}-1\right] \\
&= \mathbb{E}_{\hat{\pi}^{k+1}(\mathbf{a}|\mathbf{s})}\left[\mathcal{B}^\pi\hat{Q}^k(\mathbf{s},\mathbf{a})\right] - \underbrace{\mathbb{E}_{\pi_{\hat{Q}^k}(\mathbf{a}|\mathbf{s})}\left[\frac{\pi_{\hat{Q}^k}(\mathbf{a}|\mathbf{s})}{\hat{\pi}_\beta(\mathbf{a}|\mathbf{s})}-1\right]}_{\text{underestimation, (a)}} \\
&\quad + \sum_{\mathbf{a}}\underbrace{\left(\pi_{\hat{Q}^k}(\mathbf{a}|\mathbf{s})-\hat{\pi}^{k+1}(\mathbf{a}|\mathbf{s})\right)\frac{\pi_{\hat{Q}^k}(\mathbf{a}|\mathbf{s})}{\hat{\pi}_\beta(\mathbf{a}|\mathbf{s})}}_{\text{(b)},\ \leq\mathrm{D}_{\mathrm{TV}}(\pi_{\hat{Q}^k},\hat{\pi}^{k+1})}
\end{aligned}
$$

If (a) has a larger magnitude than (b), then the learned Q-value induces an underestimation in an iteration $k+1$, and hence, by a recursive argument, the learned Q-value underestimates the optimal Q-value. We note that by upper bounding term (b) by $\mathrm{D}_{\mathrm{TV}}(\pi_{\hat{Q}^k},\hat{\pi}^{k+1})\cdot\max_{\mathbf{a}}\frac{\pi_{\hat{Q}^k}(\mathbf{a}|\mathbf{s})}{\hat{\pi}_\beta(\mathbf{a}|\mathbf{s})}$, and writing out (a) > upper-bound on (b), we obtain the desired result.

Finally, we show that under specific choices of $\alpha_1,\cdots,\alpha_k$, the CQL backup is gap-expanding by providing a proof for Theorem 3.4

**Proof of Theorem 3.4 (CQL is gap-expanding).** For this theorem, we again first present the proof in the absence of sampling error, and then incorporate sampling error into the choice of $\alpha$. We follow the strategy of observing the Q-value update in one iteration. Recall that the expression for the Q-value iterate at iteration $k$ is given by:

$$\hat{Q}^{k+1}(\mathbf{s},\mathbf{a}) = \mathcal{B}^{\pi^k}\hat{Q}^k(\mathbf{s},\mathbf{a}) - \alpha_k \frac{\mu_k(\mathbf{a}|\mathbf{s})-\pi_\beta(\mathbf{a}|\mathbf{s})}{\pi_\beta(\mathbf{a}|\mathbf{s})}.$$

Now, the value of the policy $\mu_k(\mathbf{a}|\mathbf{s})$ under $\hat{Q}^{k+1}$ is given by:

$$\mathbb{E}_{\mathbf{a}\sim\mu_k(\mathbf{a}|\mathbf{s})}[\hat{Q}^{k+1}(\mathbf{s},\mathbf{a})] = \mathbb{E}_{\mathbf{a}\sim\mu_k(\mathbf{a}|\mathbf{s})}[\mathcal{B}^{\pi^k}\hat{Q}^k(\mathbf{s},\mathbf{a})] - \alpha_k \underbrace{\mu_k^T\left(\frac{\mu_k(\mathbf{a}|\mathbf{s})-\pi_\beta(\mathbf{a}|\mathbf{s})}{\pi_\beta(\mathbf{a}|\mathbf{s})}\right)}_{:=\hat{\Delta}^k,\ \geq 0,\ \text{by proof of Theorem 3.2.}}$$

Now, we also note that the expected amount of extra underestimation introduced at iteration $k$ under action sampled from the behavior policy $\pi_\beta(\mathbf{a}|\mathbf{s})$ is 0, as,

$$\mathbb{E}_{\mathbf{a}\sim\pi_\beta(\mathbf{a}|\mathbf{s})}[\hat{Q}^{k+1}(\mathbf{s},\mathbf{a})] = \mathbb{E}_{\mathbf{a}\sim\pi_\beta(\mathbf{a}|\mathbf{s})}[\mathcal{B}^{\pi^k}\hat{Q}^k(\mathbf{s},\mathbf{a})] - \alpha_k \underbrace{\pi_\beta^T\left(\frac{\mu_k(\mathbf{a}|\mathbf{s})-\pi_\beta(\mathbf{a}|\mathbf{s})}{\pi_\beta(\mathbf{a}|\mathbf{s})}\right)}_{=0}.$$

where the marked quantity is equal to 0 since it is equal since $\pi_\beta(\mathbf{a}|\mathbf{s})$ in the numerator cancels with the denominator, and the remaining quantity is a sum of difference between two density functions, $\sum_{\mathbf{a}}\mu_k(\mathbf{a}|\mathbf{s})-\pi_\beta(\mathbf{a}|\mathbf{s})$, which is equal to 0. Thus, we have shown that,

$$\mathbb{E}_{\pi_\beta(\mathbf{a}|\mathbf{s})}[\hat{Q}^{k+1}(\mathbf{s},\mathbf{a})]-\mathbb{E}_{\mu_k(\mathbf{a}|\mathbf{s})}[\hat{Q}^{k+1}(\mathbf{s},\mathbf{a})] = \mathbb{E}_{\pi_\beta(\mathbf{a}|\mathbf{s})}[\mathcal{B}^{\pi^k}\hat{Q}^k(\mathbf{s},\mathbf{a})]-\mathbb{E}_{\mu_k(\mathbf{a}|\mathbf{s})}[\mathcal{B}^{\pi^k}\hat{Q}^k(\mathbf{s},\mathbf{a})]-\alpha_k\hat{\Delta}^k.$$

Now subtracting the difference, $\mathbb{E}_{\pi_\beta(\mathbf{a}|\mathbf{s})}[Q^{k+1}(\mathbf{s},\mathbf{a})] - \mathbb{E}_{\mu_k(\mathbf{a}|\mathbf{s})}[Q^{k+1}(\mathbf{s},\mathbf{a})]$, computed under the tabular Q-function iterate, $Q^{k+1}$, from the previous equation, we obtain that

$$\mathbb{E}_{\mathbf{a}\sim\pi_\beta(\mathbf{a}|\mathbf{s})}[\hat{Q}^{k+1}(\mathbf{s},\mathbf{a})] - \mathbb{E}_{\pi_\beta(\mathbf{a}|\mathbf{s})}[Q^{k+1}(\mathbf{s},\mathbf{a})] = \mathbb{E}_{\mu_k(\mathbf{a}|\mathbf{s})}[\hat{Q}^{k+1}(\mathbf{s},\mathbf{a})] - \mathbb{E}_{\mu_k(\mathbf{a}|\mathbf{s})}[Q^{k+1}(\mathbf{s},\mathbf{a})]$$
$$+ (\mu_k(\mathbf{a}|\mathbf{s}) - \pi_\beta(\mathbf{a}|\mathbf{s}))^T \underbrace{\left[\mathcal{B}^{\pi^k}\left(\hat{Q}^k - Q^k\right)(\mathbf{s},\cdot)\right]}_{(a)} - \alpha_k\hat{\Delta}^k.$$

Now, by choosing $\alpha_k$, such that any positive bias introduced by the quantity $(\mu_k(\mathbf{a}|\mathbf{s}) - \pi_\beta(\mathbf{a}|\mathbf{s}))^T(a)$ is cancelled out, we obtain the following gap-expanding relationship:

$$\mathbb{E}_{\mathbf{a}\sim\pi_\beta(\mathbf{a}|\mathbf{s})}[\hat{Q}^{k+1}(\mathbf{s},\mathbf{a})] - \mathbb{E}_{\pi_\beta(\mathbf{a}|\mathbf{s})}[Q^{k+1}(\mathbf{s},\mathbf{a})] > \mathbb{E}_{\mu_k(\mathbf{a}|\mathbf{s})}[\hat{Q}^{k+1}(\mathbf{s},\mathbf{a})] - \mathbb{E}_{\mu_k(\mathbf{a}|\mathbf{s})}[Q^{k+1}(\mathbf{s},\mathbf{a})]$$

for, $\alpha_k$ satisfying,

$$\alpha_k > \max\left(\frac{(\pi_\beta(\mathbf{a}|\mathbf{s}) - \mu_k(\mathbf{a}|\mathbf{s}))^T\left[\mathcal{B}^{\pi^k}\left(\hat{Q}^k - Q^k\right)(\mathbf{s},\cdot)\right]}{\hat{\Delta}^k}, 0\right),$$

thus proving the desired result.

To avoid the dependency on the true Q-value iterate, $Q^k$, we can upper-bound $Q^k$ by $\frac{R_{\max}}{1-\gamma}$, and upper-bound $(\pi_\beta(\mathbf{a}|\mathbf{s}) - \mu_k(\mathbf{a}|\mathbf{s}))^T\mathcal{B}^{\pi^k}Q^k(\mathbf{s},\cdot)$ by $D_{\mathrm{TV}}(\pi_\beta,\mu_k)\cdot\frac{R_{\max}}{1-\gamma}$, and use this in the expression for $\alpha_k$. While this bound may be loose, it still guarantees the gap-expanding property, and we indeed empirically show the existence of this property in practice in Appendix B.

To incorporate sampling error, we can follow a similar strategy as previous proofs: the worst case overestimation due to sampling error is given by $\frac{C_{r,T,\delta}R_{\max}}{1-\gamma}$. In this case, we note that, w.h.p.,

$$\left|\hat{\mathcal{B}}^{\pi^k}\left(\hat{Q}^k - Q^k\right) - \mathcal{B}^{\pi^k}\left(\hat{Q}^k - Q^k\right)\right| \leq \frac{2\cdot C_{r,T,\delta}R_{\max}}{1-\gamma}.$$

Hence, the presence of sampling error adds $D_{\mathrm{TV}}(\hat{\pi_\beta},\mu_k)\cdot\frac{2\cdot C_{r,T,\delta}R_{\max}}{1-\gamma}$ to the value of $\alpha_k$, giving rise to the following, sufficient condition on $\alpha_k$ for the gap-expanding property:

$$\alpha_k > \max\left(\frac{(\pi_\beta(\mathbf{a}|\mathbf{s}) - \mu_k(\mathbf{a}|\mathbf{s}))^T\left[\mathcal{B}^{\pi^k}\left(\hat{Q}^k - Q^k\right)(\mathbf{s},\cdot)\right]}{\hat{\Delta}^k} + D_{\mathrm{TV}}(\hat{\pi_\beta},\mu_k)\cdot\frac{2\cdot C_{r,T,\delta}R_{\max}}{1-\gamma}, 0\right),$$

concluding the proof of this theorem.

# D   Additional Theoretical Analysis

In this section, we present a theoretical analysis of additional properties of CQL. For ease of presentation, we state and prove theorems in Appendices D.1 and D.2 in the absence of sampling error, but as discussed extensively in Appendix C, we can extend each of these results by adding extra terms induced due to sampling error.

## D.1   CQL with Linear and Non-Linear Function Approximation

**Theorem D.1.** *Assume that the Q-function is represented as a linear function of given state-action feature vectors $\mathbf{F}$, i.e., $Q(s,a) = w^T\mathbf{F}(s,a)$. Let $D = diag\left(d^{\pi_\beta}(\mathbf{s})\pi_\beta(\mathbf{a}|\mathbf{s})\right)$ denote the diagonal matrix with data density, and assume that $\mathbf{F}^T D\mathbf{F}$ is invertible. Then, the expected value of the policy under Q-value from Eqn 2 at iteration $k+1$, $\mathbb{E}_{d^{\pi_\beta}(\mathbf{a})}[\hat{V}^{k+1}(\mathbf{s})] = \mathbb{E}_{d^{\pi_\beta}(\mathbf{s}),\pi(\mathbf{a}|\mathbf{s})}[\hat{Q}^{k+1}(\mathbf{s},\mathbf{a})]$, lower-bounds the corresponding tabular value, $\mathbb{E}_{d^{\pi_\beta}(\mathbf{s})}[V^{k+1}(\mathbf{s})] = \mathbb{E}_{d^{\pi_\beta}(\mathbf{s}),\pi(\mathbf{a}|\mathbf{s})}[Q^{k+1}(\mathbf{s},\mathbf{a})]$, if*

$$\alpha_k \geq \max\left(\frac{D^T\left[\mathbf{F}\left(\mathbf{F}^T D\mathbf{F}\right)^{-1}\mathbf{F}^T - I\right]\left((\mathcal{B}^\pi\hat{Q}^k)(\mathbf{s},\mathbf{a})\right)}{D^T\left[\mathbf{F}\left(\mathbf{F}^T D\mathbf{F}\right)^{-1}\mathbf{F}^T\right]\left(D\left[\frac{\pi(\mathbf{a}|\mathbf{s})-\pi_\beta(\mathbf{a}|\mathbf{s})}{\pi_\beta(\mathbf{a}|\mathbf{s})}\right]\right)}, 0\right).$$

The choice of $\alpha_k$ in Theorem D.1 intuitively amounts to compensating for overestimation in value induced if the true value function cannot be represented in the chosen linear function class (numerator), by the potential decrease in value due to the CQL regularizer (denominator). This implies that if the actual value function can be represented in the linear function class, such that the numerator can be made 0, then **any** $\alpha > 0$ is sufficient to obtain a lower bound. We now prove the theorem.

*Proof.* In order to extend the result of Theorem 3.2 to account for function approximation, we follow the similar recipe as before. We obtain the optimal solution to the optimization problem below in the family of linearly expressible Q-functions, i.e. $\mathcal{Q} := \{\mathbf{F}w | w \in \mathbb{R}^{dim(\mathbf{F})}\}$.

$$\min_{Q \in \mathcal{Q}} \ \alpha_k \cdot \left(\mathbb{E}_{d^{\pi_\beta}(\mathbf{s}), \mu(\mathbf{a}|\mathbf{s})}[Q(\mathbf{s},\mathbf{a})] - \mathbb{E}_{d^{\pi_\beta}(\mathbf{s}), \pi_\beta(\mathbf{a}|\mathbf{s})}[Q(\mathbf{s},\mathbf{a})]\right) + \frac{1}{2}\mathbb{E}_{\mathcal{D}}\left[\left(Q(\mathbf{s},\mathbf{a}) - \mathcal{B}^\pi \hat{Q}^k(\mathbf{s},\mathbf{a})\right)^2\right].$$

By substituting $Q(\mathbf{s},\mathbf{a}) = w^T \mathbf{F}(\mathbf{s},\mathbf{a})$, and setting the derivative with respect to $w$ to be 0, we obtain,

$$\alpha \sum_{\mathbf{s},\mathbf{a}} d^{\pi_\beta}(\mathbf{s}) \cdot (\mu(\mathbf{a}|\mathbf{s}) - \pi_\beta(\mathbf{a}|\mathbf{s}))\,\mathbf{F}(\mathbf{s},\mathbf{a}) + \sum_{\mathbf{s},\mathbf{a}} d^{\pi_\beta}(\mathbf{s})\pi_\beta(\mathbf{a}|\mathbf{s})\left(Q(\mathbf{s},\mathbf{a}) - \mathcal{B}^\pi \hat{Q}^k(\mathbf{s},\mathbf{a})\right)\mathbf{F}(\mathbf{s},\mathbf{a}) = 0.$$

By re-arranging terms, and converting it to vector notation, defining $D = \mathrm{diag}(d^{\pi_\beta}(\mathbf{s})\pi_\beta(\mathbf{s}))$, and referring to the parameter $w$ at the k-iteration as $w^k$ we obtain:

$$\left(\mathbf{F}^T D\mathbf{F}\right)w^{k+1} = \underbrace{\mathbf{F}^T D\left(\mathcal{B}^\pi \hat{Q}^k\right)}_{\text{LSTD iterate}} - \underbrace{\alpha_k \mathbf{F}^T \mathrm{diag}\left[d^{\pi_\beta}(\mathbf{s}) \cdot (\mu(\mathbf{a}|\mathbf{s}) - \pi_\beta(\mathbf{a}|\mathbf{s}))\right]}_{\text{underestimation}}.$$

Now, our task is to show that the term labelled as "underestimation" is indeed negative in expectation under $\mu(\mathbf{a}|\mathbf{s})$ (This is analogous to our result in the tabular setting that shows underestimated values). In order to show this, we write out the expression for the value, under the linear weights $w^{k+1}$ at state $\mathbf{s}$, after substituting $\mu = \pi$,

$$\hat{V}^{k+1}(\mathbf{s}) := \pi(\mathbf{a}|\mathbf{s})^T \mathbf{F}w^{k+1} \tag{13}$$

$$= \pi(\mathbf{a}|\mathbf{s})^T \underbrace{\mathbf{F}\left(\mathbf{F}^T D\mathbf{F}\right)^{-1}\mathbf{F}^T D\left(\mathcal{B}^\pi \hat{Q}^k\right)}_{\text{value under LSTD-Q [32]}} - \alpha_k \pi(\mathbf{a}|\mathbf{s})^T \mathbf{F}\left(\mathbf{F}^T D\mathbf{F}\right)^{-1}\mathbf{F}^T D\left[\frac{\pi(\mathbf{a}|\mathbf{s}) - \pi_\beta(\mathbf{a}|\mathbf{s})}{\pi_\beta(\mathbf{a}|\mathbf{s})}\right].$$

$$\tag{14}$$

Now, we need to reason about the penalty term. Defining, $P_\mathbf{F} := \mathbf{F}\left(\mathbf{F}^T D\mathbf{F}\right)^{-1}\mathbf{F}^T D$ as the projection matrix onto the subspace of features $\mathbf{F}$, we need to show that the product that appears as a penalty is positive: $\pi(\mathbf{a}|\mathbf{s})^T P_\mathbf{F}\left[\frac{\pi(\mathbf{a}|\mathbf{s}) - \pi_\beta(\mathbf{a}|\mathbf{s})}{\pi_\beta(\mathbf{a}|\mathbf{s})}\right] \geq 0$. In order to show this, we compute minimum value of this product optimizing over $\pi$. If the minimum value is 0, then we are done.

Let's define $f(\pi) = \pi(\mathbf{a}|\mathbf{s})^T P_\mathbf{F}\left[\frac{\pi(\mathbf{a}|\mathbf{s}) - \pi_\beta(\mathbf{a}|\mathbf{s})}{\pi_\beta(\mathbf{a}|\mathbf{s})}\right]$, our goal is to solve for $\min_\pi f(\pi)$. Setting the derivative of $f(\pi)$ with respect to $\pi$ to be equal to 0, we obtain (including Lagrange multiplier $\eta$ that guarantees $\sum_\mathbf{a} \pi(\mathbf{a}|\mathbf{s}) = 1$,

$$\left(P_\mathbf{F} + P_\mathbf{F}^T\right)\left[\frac{\pi(\mathbf{a}|\mathbf{s})}{\pi_\beta(\mathbf{a}|\mathbf{s})}\right] = P_\mathbf{F}\vec{1} + \eta\vec{1}.$$

By solving for $\eta$ (using the condition that a density function sums to 1), we obtain that the minimum value of $f(\pi)$ occurs at a $\pi^*(\mathbf{a}|\mathbf{s})$, which satisfies the following condition,

$$\left(P_\mathbf{F} + P_\mathbf{F}^T\right)\left[\frac{\pi^*(\mathbf{a}|\mathbf{s})}{\pi_\beta(\mathbf{a}|\mathbf{s})}\right] = \left(P_\mathbf{F} + P_\mathbf{F}^T\right)\vec{1}.$$

Using this relation to compute $f$, we obtain, $f(\pi^*) = 0$, indicating that the minimum value of 0 occurs when the projected density ratio matches under the matrix $(P_\mathbf{F} + P_\mathbf{F}^T)$ is equal to the projection of a vector of ones, $\vec{1}$. Thus,

$$\forall\,\pi(\mathbf{a}|\mathbf{s}), \ f(\pi) = \pi(\mathbf{a}|\mathbf{s})^T P_\mathbf{F}\left[\frac{\pi(\mathbf{a}|\mathbf{s}) - \pi_\beta(\mathbf{a}|\mathbf{s})}{\pi_\beta(\mathbf{a}|\mathbf{s})}\right] \geq 0.$$

This means that: $\forall\,\mathbf{s}, \hat{V}^{k+1}(\mathbf{s}) \leq \hat{V}^{k+1}_{\text{LSTD-Q}}(\mathbf{s})$ given identical previous $\hat{Q}^k$ values. This result indicates, that if $\alpha_k \geq 0$, the resulting CQL value estimate with linear function approximation is guaranteed to lower-bound the value estimate obtained from a least-squares temporal difference Q-learning algorithm (which only minimizes Bellman error assuming a linear Q-function parameterization), such as LSTD-Q [32], since at each iteration, CQL induces a lower-bound with respect to the previous value iterate, whereas this underestimation is absent in LSTD-Q, and an inductive argument is applicable.

So far, we have only shown that the learned value iterate, $\hat{V}^{k+1}(\mathbf{s})$ lower-bounds the value iterate obtained from LSTD-Q, $\forall\,\mathbf{s}, \hat{V}^{k+1}(\mathbf{s}) \leq \hat{V}^{k+1}_{\text{LSTD-Q}}(\mathbf{s})$. But, our final aim is to prove a stronger result, that the learned value iterate, $\hat{V}^{k+1}$, lower bounds the exact tabular value function iterate, $V^{k+1}$, at each iteration. The reason why our current result does not guarantee this is because function approximation may induce overestimation error in the linear approximation of the Q-function.

In order to account for this change, we make a simple change: we choose $\alpha_k$ such that the resulting penalty nullifes the effect of any over-estimation caused due to the inability to fit the true value function iterate in the linear function class parameterized by $\mathbf{F}$. Formally, this means:

$$
\begin{aligned}
\mathbb{E}_{d^{\pi_\beta}(\mathbf{s})}\left[\hat{V}^{k+1}(\mathbf{s})\right] &\leq \mathbb{E}_{d^{\pi_\beta}(\mathbf{s})}\left[\hat{V}^{k+1}_{\text{LSTD-Q}}(\mathbf{s})\right] - \alpha_k \mathbb{E}_{d^{\pi_\beta}(\mathbf{s})}[f(\pi(\mathbf{a}|\mathbf{s}))] \\
&\leq \mathbb{E}_{d^{\pi_\beta}(\mathbf{s})}\left[V^{k+1}(\mathbf{s})\right] - \underbrace{\mathbb{E}_{d^{\pi_\beta}(\mathbf{s})}\left[\hat{V}^{k+1}_{\text{LSTD-Q}}(\mathbf{s}) - V^{k+1}(\mathbf{s})\right] - \alpha_k \mathbb{E}_{d^{\pi_\beta}(\mathbf{s})}[f(\pi(\mathbf{a}|\mathbf{s}))]}_{\text{choose }\alpha_k \text{ to make this negative}} \\
&\leq \mathbb{E}_{d^{\pi_\beta}(\mathbf{s})}\left[V^{k+1}(\mathbf{s})\right]
\end{aligned}
$$

And the choice of $\alpha_k$ in that case is given by:

$$
\begin{aligned}
\alpha_k &\geq \max\left(\frac{\mathbb{E}_{d^{\pi_\beta}(\mathbf{s})}\left[\hat{V}^{k+1}_{\text{LSTD-Q}}(\mathbf{s}) - V^{k+1}(\mathbf{s})\right]}{\mathbb{E}_{d^{\pi_\beta}(\mathbf{s})}[f(\pi(\mathbf{a}|\mathbf{s}))]}, 0\right) \\
&\geq \max\left(\frac{D^T\left[\mathbf{F}\left(\mathbf{F}^T D\mathbf{F}\right)^{-1}\mathbf{F}^T - I\right]\left((\mathcal{B}^\pi \hat{Q}^k)(\mathbf{s},\mathbf{a})\right)}{D^T\left[\mathbf{F}\left(\mathbf{F}^T D\mathbf{F}\right)^{-1}\mathbf{F}^T\right]\left(D\left[\frac{\pi(\mathbf{a}|\mathbf{s}) - \pi_\beta(\mathbf{a}|\mathbf{s})}{\pi_\beta(\mathbf{a}|\mathbf{s})}\right]\right)}, 0\right).
\end{aligned}
$$

Finally, we note that since this choice of $\alpha_k$ induces under-estimation in the next iterate, $\hat{V}^{k+1}$ with respect to the previous iterate, $\hat{V}^k$, for all $k \in \mathbb{N}$, by induction, we can claim that this choice of $\alpha_1, \cdots, \alpha_k$ is sufficient to make $\hat{V}^{k+1}$ lower-bound the tabular, exact value-function iterate. $V^{k+1}$, for all $k$, thus completing our proof. $\qquad\square$

We can generalize Theorem D.1 to non-linear function approximation, such as neural networks, under the standard NTK framework [25], assuming that each iteration $k$ is performed by a single step of gradient descent on Equation 2, rather than a complete minimization of this objective. As we show in Theorem D.2, CQL learns lower bounds in this case for an appropriate choice of $\alpha_k$. We will also empirically show in Appendix G that CQL can learn effective conservative Q-functions with multilayer neural networks.

**Theorem D.2** (Extension to non-linear function approximation). *Assume that the Q-function is represented by a general non-linear function approximator parameterized by $\theta$, $Q_\theta(\mathbf{s},\mathbf{a})$. let $D = diag(d^{\pi_\beta}(\mathbf{s})\pi_\beta(\mathbf{a}|\mathbf{s}))$ denote the matrix with the data density on the diagonal, and assume that $\nabla_\theta Q_\theta^T D\nabla_\theta Q_\theta$ is invertible. Then, the expected value of the policy under the Q-function obtained by taking a gradient step on Equation 2, at iteration $k+1$ lower-bounds the corresponding tabular function iterate if:*

*Proof.* Our proof strategy is to reduce the non-linear optimization problem into a linear one, with features $\mathbf{F}$ (in Theorem D.1) replaced with features given by the gradient of the current Q-function $\hat{Q}^k_\theta$ with respect to parameters $\theta$, i.e. $\nabla_\theta \hat{Q}^k$. To see, this we start by writing down the expression for

$\hat{Q}_\theta^{k+1}$ obtained via one step of gradient descent with step size $\eta$, on the objective in Equation 2.

$$\theta^{k+1} = \theta^k - \eta\alpha_k \left( \mathbb{E}_{d^{\pi_\beta}(\mathbf{s}),\mu(\mathbf{a}|\mathbf{s})} \left[ \nabla_\theta \hat{Q}^k(\mathbf{s},\mathbf{a}) \right] - \mathbb{E}_{d^{\pi_\beta}(\mathbf{s}),\pi_\beta(\mathbf{a}|\mathbf{s})} \left[ \nabla_\theta \hat{Q}^k(\mathbf{s},\mathbf{a}) \right] \right)$$
$$- \eta\mathbb{E}_{d^{\pi_\beta}(\mathbf{s}),\pi_\beta(\mathbf{a}|\mathbf{s})} \left[ \left( \hat{Q}^k - \mathcal{B}^\pi \hat{Q}^k \right)(\mathbf{s},\mathbf{a}) \cdot \nabla_\theta \hat{Q}^k(\mathbf{s},\mathbf{a}) \right].$$

Using the above equation and making an approximation linearization assumption on the non-linear Q-function, for small learning rates $\eta << 1$, as has been commonly used by prior works on the neural tangent kernel (NTK) in deep learning theory [25] in order to explain neural network learning dynamics in the infinite-width limit, we can write out the expression for the next Q-function iterate, $\hat{Q}_\theta^{k+1}$ in terms of $\hat{Q}_\theta^k$ as [2, 25]:

$$\hat{Q}_\theta^{k+1}(\mathbf{s},\mathbf{a}) \approx \hat{Q}_\theta^k(\mathbf{s},\mathbf{a}) + \left( \theta^{k+1} - \theta^k \right)^T \nabla_\theta \hat{Q}_\theta^k(\mathbf{s},\mathbf{a}) \quad \text{(under NTK assumptions)}$$
$$= \hat{Q}_\theta^k(\mathbf{s},\mathbf{a}) - \eta\alpha_k \mathbb{E}_{d^{\pi_\beta}(\mathbf{s}'),\mu(\mathbf{a}'|\mathbf{s}')} \left[ \nabla_\theta \hat{Q}^k(\mathbf{s}',\mathbf{a}')^T \nabla_\theta \hat{Q}^k(\mathbf{s},\mathbf{a}) \right]$$
$$+ \eta\alpha_k \mathbb{E}_{d^{\pi_\beta}(\mathbf{s}'),\pi_\beta(\mathbf{a}'|\mathbf{s}')} \left[ \nabla_\theta \hat{Q}^k(\mathbf{s}',\mathbf{a}')^T \nabla_\theta \hat{Q}^k(\mathbf{s},\mathbf{a}) \right]$$
$$- \eta\mathbb{E}_{d^{\pi_\beta}(\mathbf{s}'),\pi_\beta(\mathbf{a}'|\mathbf{s}')} \left[ \left( \hat{Q}^k - \mathcal{B}^\pi \hat{Q}^k \right)(\mathbf{s}',\mathbf{a}') \cdot \nabla_\theta \hat{Q}^k(\mathbf{s}',\mathbf{a}')^T \nabla_\theta \hat{Q}^k(\mathbf{s},\mathbf{a}) \right].$$

To simplify presentation, we convert into matrix notation, where we define the $|\mathcal{S}||\mathcal{A}| \times |\mathcal{S}||\mathcal{A}|$ matrix, $\mathbf{M}^k = \left( \nabla_\theta \hat{Q}^k \right)^T \nabla_\theta \hat{Q}^k$, as the neural tangent kernel matrix of the Q-function at iteration $k$. Then, the vectorized $\hat{Q}^{k+1}$ (with $\mu = \pi$) is given by,

$$\hat{Q}^{k+1} = \hat{Q}^k - \eta\alpha_k \mathbf{M}^k D \left[ \frac{\pi(\mathbf{a}|\mathbf{s}) - \pi_\beta(\mathbf{a}|\mathbf{s})}{\pi_\beta(\mathbf{a}|\mathbf{s})} \right] + \eta\mathbf{M}^k D \left( \mathcal{B}^\pi \hat{Q}^k - \hat{Q}^k \right).$$

Finally, the value of the policy is given by:

$$\hat{V}^{k+1} := \underbrace{\pi(\mathbf{a}|\mathbf{s})^T \hat{Q}^k(\mathbf{s},\mathbf{a}) + \eta\pi(\mathbf{a}|\mathbf{s})\mathbf{M}^k D \left( \mathcal{B}^\pi \hat{Q}^k - \hat{Q}^k \right)}_{\text{(a) unpenalized value}} - \underbrace{\eta\alpha_k \pi(\mathbf{a}|\mathbf{s})^T \mathbf{M}^k D \left[ \frac{\pi(\mathbf{a}|\mathbf{s}) - \pi_\beta(\mathbf{a}|\mathbf{s})}{\pi_\beta(\mathbf{a}|\mathbf{s})} \right]}_{\text{(b) penalty}}.$$

$$(15)$$

Term marked (b) in the above equation is similar to the penalty term shown in Equation 14, and by performing a similar analysis, we can show that (b) $\geq 0$. Again similar to how $\hat{V}_{\text{LSTD-Q}}^{k+1}$ appeared in Equation 14, we observe that here we obtain the value function corresponding to a regular gradient-step on the Bellman error objective. $\qquad\square$

Again similar to before, term (a) can introduce overestimation relative to the tabular counterpart, starting at $\hat{Q}^k$: $Q^{k+1} = \hat{Q}^k - \eta \left( \mathcal{B}^\pi \hat{Q}^k - \hat{Q}^k \right)$, and we can choose $\alpha_k$ to compensate for this potential increase as in the proof of Theorem D.1. As the last step, we can recurse this argument to obtain our final result, for underestimation.

### D.2 Choice of Distribution to Maximize Expected Q-Value in Equation 2

In Section 3.1, we introduced a term that maximizes Q-values under the dataset $d^{\pi_\beta}(\mathbf{s})\pi_\beta(\mathbf{a}|\mathbf{s})$ distribution when modifying Equation 1 to Equation 2. Theorem 3.2 indicates the "sufficiency" of maximizing Q-values under the dataset distribution – this guarantees a lower-bound on value. We now investigate the neccessity of this assumption: We ask the formal question: **For which other choices of $\nu(\mathbf{a}|\mathbf{s})$ for the maximization term, is the value of the policy under the learned Q-value, $\hat{Q}_\nu^{k+1}$ guaranteed to be a lower bound on the actual value of the policy?**

To recap and define notation, we restate the objective from Equation 1 below.

$$\hat{Q}^{k+1} \leftarrow \arg\min_Q \ \alpha \, \mathbb{E}_{\mathbf{s}\sim d^{\pi_\beta}(\mathbf{s}),\mathbf{a}\sim\mu(\mathbf{a}|\mathbf{s})} \left[ Q(\mathbf{s},\mathbf{a}) \right] + \frac{1}{2} \, \mathbb{E}_{\mathbf{s},\mathbf{a},\mathbf{s}'\sim\mathcal{D}} \left[ \left( Q(\mathbf{s},\mathbf{a}) - \mathcal{B}^\pi \hat{Q}^k(\mathbf{s},\mathbf{a}) \right)^2 \right]. \quad (16)$$

We define a general family of objectives from Equation 2, parameterized by a distribution $\nu$ which is chosen to maximize Q-values as shown below (CQL is a special case, with $\nu(\mathbf{a}|\mathbf{s}) = \pi_\beta(\mathbf{a}|\mathbf{s})$):

$$\hat{Q}_\nu^{k+1} \leftarrow \arg\min_Q \; \alpha \cdot \left( \mathbb{E}_{\mathbf{s}\sim d^{\pi_\beta}(\mathbf{s}), \mathbf{a}\sim\mu(\mathbf{a}|\mathbf{s})}[Q(\mathbf{s},\mathbf{a})] - \textcolor{red}{\mathbb{E}_{\mathbf{s}\sim d^{\pi_\beta}(\mathbf{s}), \mathbf{a}\sim\nu(\mathbf{a}|\mathbf{s})}[Q(\mathbf{s},\mathbf{a})]} \right)$$

$$+ \frac{1}{2} \mathbb{E}_{\mathbf{s},\mathbf{a},\mathbf{s}'\sim\mathcal{D}} \left[ \left( Q(\mathbf{s},\mathbf{a}) - \mathcal{B}^\pi \hat{Q}^k(\mathbf{s},\mathbf{a}) \right)^2 \right]. \quad (\nu\text{-CQL}) \quad (17)$$

In order to answer our question, we prove the following result:

**Theorem D.3** (Necessity of maximizing Q-values under $\pi_\beta(\mathbf{a}|\mathbf{s})$.)**.** *For any policy $\pi(\mathbf{a}|\mathbf{s})$, any $\alpha > 0$, and for all $k > 0$, the value of the policy., $\hat{V}_\nu^{k+1}$ under Q-function iterates from $\nu - CQL$, $\hat{Q}_\nu^{k+1}(\mathbf{s},\mathbf{a})$ is guaranteed to be a lower bound on the exact value iterate, $\hat{V}^{k+1}$, only if $\nu(\mathbf{a}|\mathbf{s}) = \pi_\beta(\mathbf{a}|\mathbf{s})$.*

*Proof.* We start by noting the parametric form of the resulting tabular Q-value iterate:

$$\hat{Q}_\nu^{k+1}(\mathbf{s},\mathbf{a}) = \mathcal{B}^\pi \hat{Q}_\nu^k(\mathbf{s},\mathbf{a}) - \alpha_k \frac{\mu(\mathbf{a}|\mathbf{s}) - \nu(\mathbf{a}|\mathbf{s})}{\pi_\beta(\mathbf{a}|\mathbf{s})}. \quad (18)$$

The value of the policy under this Q-value iterate, when distribution $\mu$ is chosen to be the target policy $\pi(\mathbf{a}|\mathbf{s})$ i.e. $\mu(\mathbf{a}|\mathbf{s}) = \pi(\mathbf{a}|\mathbf{s})$ is given by:

$$\hat{V}_\nu^{k+1}(\mathbf{s}) := \mathbb{E}_{\mathbf{a}\sim\pi(\mathbf{a}|\mathbf{s})} \left[ \hat{Q}_\nu^{k+1}(\mathbf{s},\mathbf{a}) \right] = \mathbb{E}_{\mathbf{a}\sim\pi(\mathbf{a}|\mathbf{s})} \left[ \mathcal{B}^\pi \hat{Q}_\nu^k(\mathbf{s},\mathbf{a}) \right] - \alpha_k \, \pi(\mathbf{a}|\mathbf{s})^T \left( \frac{\pi(\mathbf{a}|\mathbf{s}) - \nu(\mathbf{a}|\mathbf{s})}{\pi_\beta(\mathbf{a}|\mathbf{s})} \right). \quad (19)$$

We are interested in conditions on $\nu(\mathbf{a}|\mathbf{s})$ such that the penalty term in the above equation is positive. It is clear that choosing $\nu(\mathbf{a}|\mathbf{s}) = \pi_\beta(\mathbf{a}|\mathbf{s})$ returns a policy that satisfies the requirement, as shown in the proof for Theorem 3.2. In order to obtain other choices of $\nu(\mathbf{a}|\mathbf{s})$ that guarantees a lower bound for all possible choices of $\pi(\mathbf{a}|\mathbf{s})$, we solve the following concave-convex maxmin optimization problem, that computes a $\nu(\mathbf{a}|\mathbf{s})$ for which a lower-bound is guaranteed for *all* choices of $\mu(\mathbf{a}|\mathbf{s})$:

$$\max_{\nu(\mathbf{a}|\mathbf{s})} \min_{\pi(\mathbf{a}|\mathbf{s})} \; \sum_{\mathbf{a}} \pi(\mathbf{a}|\mathbf{s}) \cdot \left( \frac{\pi(\mathbf{a}|\mathbf{s}) - \nu(\mathbf{a}|\mathbf{s})}{\pi_\beta(\mathbf{a}|\mathbf{s})} \right)$$

$$\text{s.t.} \quad \sum_{\mathbf{a}} \pi(\mathbf{a}|\mathbf{s}) = 1, \; \sum_{\mathbf{a}} \nu(\mathbf{a}|\mathbf{s}) = 1, \; \nu(\mathbf{a}|\mathbf{s}) \geq 0, \; \pi(\mathbf{a}|\mathbf{s}) \geq 0.$$

We first solve the inner minimization over $\pi(\mathbf{a}|\mathbf{s})$ for a fixed $\nu(\mathbf{a}|\mathbf{s})$, by writing out the Lagrangian and setting the gradient of the Lagrangian to be 0, we obtain:

$$\forall \mathbf{a}, \; 2 \cdot \frac{\pi^*(\mathbf{a}|\mathbf{s})}{\pi_\beta(\mathbf{a}|\mathbf{s})} - \frac{\nu(\mathbf{a}|\mathbf{s})}{\pi_\beta(\mathbf{a}|\mathbf{s})} - \zeta(\mathbf{a}|\mathbf{s}) + \eta = 0,$$

where $\zeta(\mathbf{a}|\mathbf{s})$ is the Lagrange dual variable for the positivity constraints on $\pi(\mathbf{a}|\mathbf{s})$, and $\eta$ is the Lagrange dual variable for the normalization constraint on $\pi$. If $\pi(\mathbf{a}|\mathbf{s})$ is full support (for example, when it is chosen to be a Boltzmann policy), KKT conditions imply that, $\zeta(\mathbf{a}|\mathbf{s}) = 0$, and computing $\eta$ by summing up over actions, $\mathbf{a}$, the optimal choice of $\pi$ for the inner minimization is given by:

$$\pi^*(\mathbf{a}|\mathbf{s}) = \frac{1}{2}\nu(\mathbf{a}|\mathbf{s}) + \frac{1}{2}\pi_\beta(\mathbf{a}|\mathbf{s}). \quad (20)$$

Now, plugging Equation 20 in the original optimization problem, we obtain the following optimization over only $\nu(\mathbf{a}|\mathbf{s})$:

$$\max_{\nu(\mathbf{a}|\mathbf{s})} \sum_{\mathbf{a}} \pi_\beta(\mathbf{a}|\mathbf{s}) \cdot \left( \frac{1}{2} - \frac{\nu(\mathbf{a}|\mathbf{s})}{2\pi_\beta(\mathbf{a}|\mathbf{s})} \right) \cdot \left( \frac{1}{2} + \frac{\nu(\mathbf{a}|\mathbf{s})}{2\pi_\beta(\mathbf{a}|\mathbf{s})} \right) \quad \text{s.t.} \quad \sum_{\mathbf{a}} \nu(\mathbf{a}|\mathbf{s}) = 1, \; \nu(\mathbf{a}|\mathbf{s}) \geq 0. \quad (21)$$

Solving this optimization, we find that the optimal distribution, $\nu(\mathbf{a}|\mathbf{s})$ is equal to $\pi_\beta(\mathbf{a}|\mathbf{s})$. and the optimal value of penalty, which is also the objective for the problem above is equal to 0. Since we are maximizing over $\nu$, this indicates for other choices of $\nu \neq \pi_\beta$, we can find a $\pi$ so that the penalty is negative, and hence a lower-bound is not guaranteed. Therefore, we find that with a worst case choice of $\pi(\mathbf{a}|\mathbf{s})$, a lower bound can only be guaranteed only if $\nu(\mathbf{a}|\mathbf{s}) = \pi_\beta(\mathbf{a}|\mathbf{s})$. This justifies the necessity of $\pi_\beta(\mathbf{a}|\mathbf{s})$ for maximizing Q-values in Equation 2. The above analysis doesn't take into account the effect of function approximation or sampling error. We can, however, generalize this result to those settings, by following a similar strategy of appropriately choosing $\alpha_k$, as previously utilized in Theorem D.1. $\qquad\square$

## D.3 CQL with Empirical Dataset Distributions

The results in Sections 3.1 and 3.2 account for sampling error due to the finite size of the dataset $\mathcal{D}$. In our practical implementation as well, we optimize a sample-based version of Equation 2, as shown below:

$$\hat{Q}^{k+1} \leftarrow \arg\min_{Q} \ \alpha \cdot \left( \sum_{\mathbf{s} \in \mathcal{D}} \mathbb{E}_{\mathbf{a} \sim \mu(\mathbf{a}|\mathbf{s})} \left[ Q(\mathbf{s}, \mathbf{a}) \right] - \sum_{\mathbf{s} \in \mathcal{D}} \mathbb{E}_{\mathbf{a} \sim \pi_\beta(\mathbf{a}|\mathbf{s})} \left[ Q(\mathbf{s}, \mathbf{a}) \right] \right)$$
$$+ \frac{1}{2|\mathcal{D}|} \sum_{\mathbf{s},\mathbf{a},\mathbf{s}' \in \mathcal{D}} \left[ \left( Q(\mathbf{s}, \mathbf{a}) - \hat{\mathcal{B}}^\pi \hat{Q}^k(\mathbf{s}, \mathbf{a}) \right)^2 \right], \quad (22)$$

where $\hat{\mathcal{B}}^\pi$ denotes the "empirical" Bellman operator computed using samples in $\mathcal{D}$ as follows:

$$\forall \mathbf{s}, \mathbf{a} \in \mathcal{D}, \quad \left( \hat{\mathcal{B}}^\pi \hat{Q}^k \right)(\mathbf{s}, \mathbf{a}) = r + \gamma \sum_{\mathbf{s}'} \hat{T}(\mathbf{s}'|\mathbf{s}, \mathbf{a}) \mathbb{E}_{\mathbf{a}' \sim \pi(\mathbf{a}'|\mathbf{s}')} \left[ \hat{Q}^k(\mathbf{s}', \mathbf{a}') \right], \quad (23)$$

where $r$ is the empirical average reward obtained in the dataset when executing an action $\mathbf{a}$ at state $\mathbf{s}$, i.e. $r = \frac{1}{|\mathcal{D}(\mathbf{s},\mathbf{a})|} \sum_{\mathbf{s}_i, \mathbf{a}_i in \mathcal{D}} \mathbf{1}_{\mathbf{s}_i = \mathbf{s}, \mathbf{a}_i = \mathbf{a}} \cdot r(\mathbf{s}, \mathbf{a})$, and $\hat{T}(\mathbf{s}'|\mathbf{s}, \mathbf{a})$ is the empirical transition matrix. Note that expectation under $\pi(\mathbf{a}|\mathbf{s})$ can be computed exactly, since it does not depend on the dataset. The empirical Bellman operator can take higher values as compared to the actual Bellman operator, $\mathcal{B}^\pi$, for instance, in an MDP with stochastic dynamics, where $\mathcal{D}$ may not contain transitions to all possible next-states $\mathbf{s}'$ that can be reached by executing action $\mathbf{a}$ at state $\mathbf{s}$, and only contains an optimistic transition.

We next show how the CQL lower bound result (Theorem 3.2) can be modified to guarantee a lower bound even in this presence of sampling error. To note this, following prior work [26, 46], we assume concentration properties of the reward function and the transition dynamics:

**Assumption D.1.** $\forall \, \mathbf{s}, \mathbf{a} \in \mathcal{D}$, *the following relationships hold with high probability,* $\geq 1 - \delta$

$$|r - r(\mathbf{s}, \mathbf{a})| \leq \frac{C_{r,\delta}}{\sqrt{|\mathcal{D}(\mathbf{s},\mathbf{a})|}}, \quad ||\hat{T}(\mathbf{s}'|\mathbf{s}, \mathbf{a}) - T(\mathbf{s}'|\mathbf{s}, \mathbf{a})||_1 \leq \frac{C_{T,\delta}}{\sqrt{|\mathcal{D}(\mathbf{s},\mathbf{a})|}}.$$

Under this assumption, the difference between the empirical Bellman operator and the actual Bellman operator can be bounded:

$$\left| \left( \hat{\mathcal{B}}^\pi \hat{Q}^k \right) - \left( \mathcal{B}^\pi \hat{Q}^k \right) \right| = \left| (r - r(\mathbf{s}, \mathbf{a})) + \gamma \sum_{\mathbf{s}'} \left( \hat{T}(\mathbf{s}'|\mathbf{s}, \mathbf{a}) - T(\mathbf{s}'|\mathbf{s}, \mathbf{a}) \right) \mathbb{E}_{\pi(\mathbf{a}'|\mathbf{s}')} \left[ \hat{Q}^k(\mathbf{s}', \mathbf{a}') \right] \right|$$

$$\leq |r - r(\mathbf{s}, \mathbf{a})| + \gamma \left| \sum_{\mathbf{s}'} \left( \hat{T}(\mathbf{s}'|\mathbf{s}, \mathbf{a}) - T(\mathbf{s}'|\mathbf{s}, \mathbf{a}) \right) \mathbb{E}_{\pi(\mathbf{a}'|\mathbf{s}')} \left[ \hat{Q}^k(\mathbf{s}', \mathbf{a}') \right] \right|$$

$$\leq \frac{C_{r,\delta} + \gamma C_{T,\delta} 2 R_{\max}/(1 - \gamma)}{\sqrt{|\mathcal{D}(\mathbf{s},\mathbf{a})|}}.$$

This gives us an expression to bound the potential overestimation that can occur due to sampling error, as a function of a constant, $C_{r,T,\delta}$ that can be expressed as a function of $C_{r,\delta}$ and $C_{T,\delta}$, and depends on $\delta$ via a $\sqrt{\log(1/\delta)}$ dependency. This is similar to how prior works have bounded the sampling error due to an empirical Bellman operator [46, 26].

## D.4 Safe Policy Improvement Guarantee for CQL

In this section, we prove a safe policy improvement guarantee for CQL (and this analysis is also applicable in general to policy constraint methods with appropriate choices of constraints as we will show). We define the empirical MDP, $\hat{M}$ as the MDP formed by the transitions in the replay buffer, $\hat{M} = \{\mathbf{s}, \mathbf{a}, r, \mathbf{s}' \in \mathcal{D}\}$, and let $J(\pi, \hat{M})$ denote the return of a policy $\pi(\mathbf{a}|\mathbf{s})$ in MDP $\hat{M}$. Our goal is to show that $J(\pi, M) \geq J(\hat{\pi}_\beta, M) - \varepsilon$, with high probability, where $\varepsilon$ is a small constant. We start by proving that CQL optimizes a penalized RL objective in the empirical MDP, $\hat{M}$.

**Lemma D.3.1.** *Let $\hat{Q}^\pi$ be the fixed point of Equation 2, then $\pi^*(\mathbf{a}|\mathbf{s}) := \arg\max_\pi \mathbb{E}_{\mathbf{s}\sim\rho(\mathbf{s})}[\hat{V}^\pi(\mathbf{s})]$ is equivalently obtained by solving:*

$$\pi^*(\mathbf{a}|\mathbf{s}) \leftarrow \arg\max_\pi \ J(\pi, \hat{M}) - \alpha\frac{1}{1-\gamma}\mathbb{E}_{\mathbf{s}\sim d^\pi_{\hat{M}}(\mathbf{s})}\left[D_{CQL}(\pi, \hat{\pi}_\beta)(\mathbf{s})\right], \tag{24}$$

*where $D_{CQL}(\pi, \hat{\pi}_\beta)(\mathbf{s}) := \sum_\mathbf{a} \pi(\mathbf{a}|\mathbf{s}) \cdot \left(\frac{\pi(\mathbf{a}|\mathbf{s})}{\hat{\pi}_\beta(\mathbf{a}|\mathbf{s})} - 1\right)$.*

*Proof.* $\hat{Q}^\pi$ is obtained by solving a recursive Bellman fixed point equation in the empirical MDP $\hat{M}$, with an altered reward, $r(\mathbf{s},\mathbf{a}) - \alpha\left[\frac{\pi(\mathbf{a}|\mathbf{s})}{\hat{\pi}_\beta(\mathbf{a}|\mathbf{s})} - 1\right]$, hence the optimal policy $\pi^*(\mathbf{a}|\mathbf{s})$ obtained by optimizing the value under the CQL Q-function equivalently is characterized via Equation 24. $\square$

Now, our goal is to relate the performance of $\pi^*(\mathbf{a}|\mathbf{s})$ in the *actual* MDP, $M$, to the performance of the behavior policy, $\pi_\beta(\mathbf{a}|\mathbf{s})$. To this end, we prove the following theorem:

**Theorem D.4.** *Let $\pi^*(\mathbf{a}|\mathbf{s})$ be the policy obtained by optimizing Equation 24. Then the performance of $\pi^*(\mathbf{a}|\mathbf{s})$ in the actual MDP $M$ satisfies,*

$$J(\pi^*, M) \geq J(\hat{\pi}_\beta, M) - 2\left(\frac{C_{r,\delta}}{1-\gamma} + \frac{\gamma R_{\max}C_{T,\delta}}{(1-\gamma)^2}\right)\mathbb{E}_{\mathbf{s}\sim d^{\pi^*}_{\hat{M}}(\mathbf{s})}\left[\frac{\sqrt{|\mathcal{A}|}}{\sqrt{|\mathcal{D}(\mathbf{s})|}}\sqrt{D_{CQL}(\pi^*, \hat{\pi}_\beta)(\mathbf{s}) + 1}\right]$$

$$+ \alpha\frac{1}{1-\gamma}\mathbb{E}_{\mathbf{s}\sim d^{\pi^*}_{\hat{M}}(\mathbf{s})}\left[D_{CQL}(\pi^*, \hat{\pi}_\beta)(\mathbf{s})\right]. \tag{25}$$

*Proof.* The proof for this statement is divided into two parts. The first part involves relating the return of $\pi^*(\mathbf{a}|\mathbf{s})$ in MDP $\hat{M}$ with the return of $\hat{\pi}_\beta$ in MDP $\hat{M}$. Since, $\pi^*(\mathbf{a}|\mathbf{s})$ optimizes Equation 24, we can relate $J(\pi^*, \hat{M})$ and $J(\hat{\pi}_\beta, \hat{M})$ as:

$$J(\pi^*, \hat{M}) - \alpha\mathbb{E}_{\mathbf{s}\sim d^\pi_{\hat{M}}(\mathbf{s})}\left[D_{CQL}(\pi^*, \pi_\beta)(\mathbf{s})\right] \geq J(\hat{\pi}_\beta, \hat{M}) - 0 = J(\hat{\pi}_\beta, \hat{M}).$$

The next step involves using concentration inequalities to upper and lower bound $J(\pi^*, \hat{M})$ and $J(\pi^*, M)$ and the corresponding difference for the behavior policy. In order to do so, we prove the following lemma, that relates $J(\pi, M)$ and $J(\pi, \hat{M})$ for an arbitrary policy $\pi$. We use this lemma to then obtain the proof for the above theorem. $\square$

**Lemma D.4.1.** *For any MDP $M$, an empirical MDP $\hat{M}$ generated by sampling actions according to the behavior policy $\hat{\pi}_\beta(\mathbf{a}|\mathbf{s})$ and a given policy $\pi$,*

$$\left|J(\pi, \hat{M}) - J(\pi, M)\right| \leq \left(\frac{C_{r,\delta}}{1-\gamma} + \frac{\gamma R_{\max}C_{T,\delta}}{(1-\gamma)^2}\right)\mathbb{E}_{\mathbf{s}\sim d^\pi_{\hat{M}}(\mathbf{s})}\left[\frac{\sqrt{|\mathcal{A}|}}{|\mathcal{D}(\mathbf{s})|}\sqrt{D_{CQL}(\pi, \hat{\pi}_\beta)(\mathbf{s}) + 1}\right].$$

*Proof.* To prove this, we first use the triangle inequality to clearly separate reward and transition dynamics contributions in the expected return.

$$\left|J(\pi, \hat{M}) - J(\pi, M)\right| = \frac{1}{1-\gamma}\left|\sum_{\mathbf{s},\mathbf{a}} d^\pi_{\hat{M}}(\mathbf{s})\pi(\mathbf{a}|\mathbf{s})r_{\hat{M}}(\mathbf{s},\mathbf{a}) - \sum_{\mathbf{s},\mathbf{a}} d^\pi_M(\mathbf{s})\pi(\mathbf{a}|\mathbf{s})r_M(\mathbf{s},\mathbf{a})\right| \tag{26}$$

$$\leq \frac{1}{1-\gamma}\left|\sum_{\mathbf{s},\mathbf{a}} d^\pi_{\hat{M}}(\mathbf{s})\underbrace{\left[\pi(\mathbf{a}|\mathbf{s})(r_{\hat{M}}(\mathbf{s},\mathbf{a}) - r_M(\mathbf{s},\mathbf{a}))\right]}_{:=\Delta_1(\mathbf{s})}\right| + \frac{1}{1-\gamma}\left|\sum_{\mathbf{s},\mathbf{a}}\left(d^\pi_{\hat{M}}(\mathbf{s}) - d^\pi_M(\mathbf{s})\right)\pi(\mathbf{a}|\mathbf{s})r_M(\mathbf{s},\mathbf{a})\right|$$

$$\tag{27}$$

We first use concentration inequalities to upper bound $\Delta_1(\mathbf{s})$. Note that under concentration assumptions, and by also using the fact that $\mathbb{E}[\Delta_1(\mathbf{s})] = 0$ (in the limit of infinite data), we get:

$$|\Delta_1(\mathbf{s})| \leq \sum_\mathbf{a} \pi(\mathbf{a}|\mathbf{s})|r_{\hat{M}}(\mathbf{s},\mathbf{a}) - r_M(\mathbf{s},\mathbf{a})| \leq \sum_\mathbf{a} \pi(\mathbf{a}|\mathbf{s})\frac{C_{r,\delta}}{\sqrt{|\mathcal{D}(\mathbf{s})| \cdot |\mathcal{D}(\mathbf{a}|\mathbf{s})|}} = \frac{C_{r,\delta}}{\sqrt{|\mathcal{D}(\mathbf{s})|}}\sum_\mathbf{a}\frac{\pi(\mathbf{a}|\mathbf{s})}{\sqrt{\hat{\pi}_\beta(\mathbf{a}|\mathbf{s})}},$$

where the last step follows from the fact that $|\mathcal{D}(\mathbf{s}, \mathbf{a})| = |\mathcal{D}(\mathbf{s})| \cdot \hat{\pi}_\beta(\mathbf{a}|\mathbf{s})$.

Next we bound the second term. We first note that if we can bound $||d^\pi_{\hat{M}} - d^\pi_M||_1$, (i.e., the total variation between the marginal state distributions in $\hat{M}$ and $M$, then we are done, since $|r_M(\mathbf{s}, \mathbf{a})| \leq R_{\max}$ and $\pi(\mathbf{a}|\mathbf{s}) \leq 1$, and hence we bound the second term effectively. We use an analysis similar to Achiam et al. [1] to obtain this total variation bound. Define, $G = (I - \gamma P^\pi_M)^{-1}$ and $\bar{G} = (I - \gamma P^\pi_{\hat{M}})^{-1}$ and let $\Delta = P^\pi_M - P^\pi_{\hat{M}}$. Then, we can write:

$$d^\pi_{\hat{M}} - d^\pi_M = (1 - \gamma)(\bar{G} - G)\rho,$$

where $\rho(\mathbf{s})$ is the initial state distribution, which is assumed to be the same for both the MDPs. Then, using Equation 21 from Achiam et al. [1], we can simplify this to obtain,

$$d^\pi_M - d^\pi_{\hat{M}} = (1 - \gamma)\gamma G\Delta\bar{G}\mu = \gamma\bar{G}\Delta d^\pi_{\hat{M}}$$

Now, following steps similar to proof of Lemma 3 in Achiam et al. [1] we obtain,

$$||\Delta d^\pi_M||_1 = \sum_{\mathbf{s}'} \left| \sum_{\mathbf{s}} \Delta(\mathbf{s}'|\mathbf{s}) d^\pi_M(\mathbf{s}) \right| \leq \sum_{\mathbf{s}, \mathbf{s}'} |\Delta(\mathbf{s}'|\mathbf{s})| \, d^\pi_{\hat{M}}$$

$$= \sum_{\mathbf{s}, \mathbf{s}'} \left| \sum_{\mathbf{a}} \left( P_{\hat{M}}(\mathbf{s}'|\mathbf{s}, \mathbf{a}) - P_M(\mathbf{s}'|\mathbf{s}, \mathbf{a}) \right) \pi(\mathbf{a}|\mathbf{s}) \right| d^\pi_{\hat{M}}(\mathbf{s})$$

$$\leq \sum_{\mathbf{s}, \mathbf{a}} ||P_{\hat{M}}(\cdot|\mathbf{s}, \mathbf{a}) - P_M(\cdot|\mathbf{s}, \mathbf{a})||_1 \pi(\mathbf{a}|\mathbf{s}) d^\pi_{\hat{M}}(\mathbf{s})$$

$$\leq \sum_{\mathbf{s}} d^\pi_{\hat{M}}(\mathbf{s}) \frac{C_{T,\delta}}{\sqrt{|\mathcal{D}(\mathbf{s})|}} \sum_{\mathbf{a}} \frac{\pi(\mathbf{a}|\mathbf{s})}{\sqrt{\hat{\pi}_\beta(\mathbf{a}|\mathbf{s})}}.$$

Hence, we can bound the second term by:

$$\left| \sum_{\mathbf{s}} \left( d^\pi_{\hat{M}}(\mathbf{s}) - d^\pi_M(\mathbf{s}) \right) \pi(\mathbf{a}|\mathbf{s}) r_M(\mathbf{s}, \mathbf{a}) \right| \leq \frac{\gamma C_{T,\delta} R_{\max}}{(1 - \gamma)} \sum_{\mathbf{s}} d^\pi_{\hat{M}}(\mathbf{s}) \frac{1}{\sqrt{|\mathcal{D}(\mathbf{s})|}} \sum_{\mathbf{a}} \frac{\pi(\mathbf{a}|\mathbf{s})}{\sqrt{\hat{\pi}_\beta(\mathbf{a}|\mathbf{s})}}.$$

To finally obtain $D_{\mathrm{CQL}}(\pi, \pi_\beta)(\mathbf{s})$ in the bound, let $\alpha(\mathbf{s}, \mathbf{a}) := \frac{\pi(\mathbf{a}|\mathbf{s})}{\sqrt{\hat{\pi}_\beta(\mathbf{a}|\mathbf{s})}}$. Then, we can write $D_{\mathrm{CQL}}(\pi, \hat{\pi}_\beta)(\mathbf{s})$ as follows:

$$D_{\mathrm{CQL}}(\mathbf{s}) = \sum_{\mathbf{a}} \frac{\pi(\mathbf{a}|\mathbf{s})^2}{\hat{\pi}_\beta(\mathbf{a}|\mathbf{s})} - 1$$

$$\implies D_{\mathrm{CQL}}(\mathbf{s}) + 1 = \sum_{\mathbf{a}} \alpha(\mathbf{s}, \mathbf{a})^2$$

$$\implies D_{\mathrm{CQL}}(\mathbf{s}) + 1 \leq \left( \sum_{\mathbf{a}} \alpha(\mathbf{s}, \mathbf{a}) \right)^2 \leq |\mathcal{A}| \left( D_{\mathrm{CQL}}(\mathbf{s}) + 1 \right).$$

Combining these arguments together, we obtain the following upper bound on $|J(\pi, M) - J(\pi, \hat{M})|$,

$$\left| J(\pi, \hat{M}) - J(\pi, M) \right| \leq \left( \frac{C_{r,\delta}}{1 - \gamma} + \frac{\gamma R_{\max} C_{T,\delta}}{(1 - \gamma)^2} \right) \mathbb{E}_{\mathbf{s} \sim d^\pi_{\hat{M}}(\mathbf{s})} \left[ \frac{\sqrt{|\mathcal{A}|}}{\sqrt{|\mathcal{D}(\mathbf{s})|}} \sqrt{D_{\mathrm{CQL}}(\pi, \hat{\pi}_\beta)(\mathbf{s}) + 1} \right].$$

$\square$

The proof of Theorem D.4 is then completed by using the above Lemma for bounding the sampling error for $\pi^*$ and then upper bounding the sampling error for $\hat{\pi}_\beta$ by the corresponding sampling error for $\pi^*$, hence giving us a factor of 2 on the sampling error term in Theorem D.4. To see why this is mathematically correct, note that $D_{\mathrm{CQL}}(\hat{\pi}_\beta, \hat{\pi}_\beta)(\mathbf{s}) = 0$, hence $\sqrt{D_{\mathrm{CQL}}(\pi^*, \hat{\pi}_\beta)(\mathbf{s}) + 1} \geq \sqrt{D_{\mathrm{CQL}}(\hat{\pi}_\beta, \hat{\pi}_\beta)(\mathbf{s}) + 1}$, which means the sampling error term for $\pi^*$ pointwise upper bounds the sampling error for $\hat{\pi}_\beta$, which justifies the factor of 2.

# E  Extended Related Work and Connections to Prior Methods

In this section, we discuss related works to supplement Section 5. Specifically, we discuss the relationships between CQL and uncertainty estimation and policy-constraint methods.

**Relationship to uncertainty estimation in offline RL.** A number of prior approaches to offline RL estimate some sort of epistemic uncertainty to determine the trustworthiness of a Q-value prediction [30, 15, 3, 34]. The policy is then optimized with respect to lower-confidence estimates derived using the uncertainty metric. However, it has been empirically noted that uncertainty-based methods are not sufficient to prevent against OOD actions [15, 30] in and of themselves, are often augmented with policy constraints due to the inability to estimate tight and calibrated uncertainty sets. Such loose or uncalibrated uncertainty sets are still effective in providing exploratory behavior in standard, online RL [47, 46], where these methods were originally developed. However, offline RL places high demands on the fidelity of such sets [34], making it hard to directly use these methods.

**How does CQL relate to prior uncertainty estimation methods?** Typical uncertainty estimation methods rely on learning a pointwise upper bound on the Q-function that depends on epistemic uncertainty [26, 46] and these upper-confidence bound values are then used for exploration. In the context of offline RL, this means learning a pointwise lower-bound on the Q-function. We show in Section 3.1 that, with a naïve choice of regularizer (Equation 1), we can learn a uniform lower-bound on the Q-function, however, we then showed that we can improve this bound since the value of the policy is the primary quantity of interest that needs to be lower-bounded. This implies that CQL strengthens the popular practice of point-wise lower-bounds made by uncertainty estimation methods.

**Can we make CQL dependent on uncertainty?** We can slightly modify CQL to make it be account for epistemic uncertainty under certain statistical concentration assumptions. Typical uncertainty estimation methods in RL [47, 26] assume the applicability of concentration inequalities (for example, by making sub-Gaussian assumptions on the reward and dynamics), to obtain upper or lower-confidence bounds and the canonical amount of over- (under-) estimation is usually given by, $\mathcal{O}\left(\frac{1}{\sqrt{n(\mathbf{s},\mathbf{a})}}\right)$, where $n(\mathbf{s},\mathbf{a})$ is the number of times a state-action pair $(\mathbf{s},\mathbf{a})$ is observed in the dataset. We can incorporate such behavior in CQL by modifying Equation 3 to update Bellman error weighted by the cardinality of the dataset, $|\mathcal{D}|$, which gives rise to the following effective Q-function update in the tabular setting, without function approximation:

$$\hat{Q}^{k+1}(\mathbf{s},\mathbf{a}) = \mathcal{B}^\pi \hat{Q}^k(\mathbf{s},\mathbf{a}) - \alpha \frac{\mu(\mathbf{a}|\mathbf{s}) - \pi_\beta(\mathbf{a}|\mathbf{s})}{n(\mathbf{s},\mathbf{a})} \to \mathcal{B}^\pi \hat{Q}^k(\mathbf{s},\mathbf{a}) \text{ as } n(\mathbf{s},\mathbf{a}) \to \infty.$$

In the limit of infinite data, i.e. $n(\mathbf{s},\mathbf{a}) \to \infty$, we find that the amount of underestimation tends to 0. When only a finite-sized dataset is provided, i.e. $n(\mathbf{s},\mathbf{a}) < N$, for some $N$, we observe that by making certain assumptions, previously used in prior work [26, 47] on the concentration properties of the reward value, $r(\mathbf{s},\mathbf{a})$ and the dynamics function, $T(\mathbf{s}'|\mathbf{s},\mathbf{a})$, such as follows:

$$||\hat{r}(\mathbf{s},\mathbf{a}) - r(\mathbf{s},\mathbf{a})|| \le \frac{C_r}{\sqrt{n(\mathbf{s},\mathbf{a})}} \quad \text{and} \quad ||\hat{T}(\mathbf{s}'|\mathbf{s},\mathbf{a}) - T(\mathbf{s}'|\mathbf{s},\mathbf{a})|| \le \frac{C_T}{\sqrt{n(\mathbf{s},\mathbf{a})}},$$

where $C_r$ and $C_T$ are constants, that depend on the concentration properties of the MDP, and by appropriately choosing $\alpha$, i.e. $\alpha = \Omega(n(\mathbf{s},\mathbf{a}))$, such that the learned Q-function still lower-bounds the actual Q-function (by nullifying the possible overestimation that appears due to finite samples), we are still guaranteed a lower bound.

# F  Additional Experimental Setup and Implementation Details

In this section, we discuss some additional implementation details related to our method. As discussed in Section 4, CQL can be implemented as either a Q-learning or an actor-critic method. For our experiments on D4RL benchmarks [12], we implemented CQL on top of soft actor-critic (SAC) [19], and for experiments on discrete-action Atari tasks, we implemented CQL on top of QR-DQN [8]. We experimented with two ways of implementing CQL, first with a fixed $\alpha$, where we chose $\alpha = 5.0$, and second with a varying $\alpha$ chosen via dual gradient-descent. The latter formulation automates the

choice of $\alpha$ by introducing a "budget" parameter, $\tau$, as shown below:

$$\min_Q \max_{\alpha \geq 0} \alpha \left( \mathbb{E}_{\mathbf{s} \sim d^{\pi_\beta}(\mathbf{s})} \left[ \log \sum_{\mathbf{a}} \exp(Q(\mathbf{s}, \mathbf{a})) - \mathbb{E}_{\mathbf{a} \sim \pi_\beta(\mathbf{a}|\mathbf{s})} [Q(\mathbf{s}, \mathbf{a})] \right] - \tau \right) + \frac{1}{2} \mathbb{E}_{\mathbf{s}, \mathbf{a}, \mathbf{s}' \sim \mathcal{D}} \left[ \left( Q - \mathcal{B}^{\pi_k} \hat{Q}^k \right)^2 \right].$$
(28)

Equation 28 implies that if the expected difference in Q-values is less than the specified threshold $\tau$, $\alpha$ will adjust to be close to 0, whereas if the difference in Q-values is higher than the specified threshold, $\tau$, then $\alpha$ is likely to take on high values, and thus more aggressively penalize Q-values. We refer to this version as CQL-Lagrange, and we found that this version drastically outperforms the fixed $\alpha$ version on the more complex AntMazes.

**Choice of $\alpha$.** Our experiments in Section 6 use either a fixed penalty value $\alpha = 5.0$ or the Lagrange version to automatically tune $\alpha$ during training. For our experiments, across D4RL Gym MuJoCo domains, we choose a fixed value at $\alpha = 5.0$. For the other D4RL domains (Franka Kitchen and Adroit), we chose $\tau = 5.0$. And for our Atari experiments, we used a fixed penalty, with $\alpha = 1.0$ chosen uniformly for Table 3 (with 10% data), $\alpha = 4.0$ chosen uniformly for Table 3 (with 1% data), and $\alpha = 0.5$ for Figure 1.

**Computing $\log \sum_{\mathbf{a}} \exp(Q(\mathbf{s}, \mathbf{a}))$.** CQL($\mathcal{H}$) uses log-sum-exp in the objective for training the Q-function (Equation 4). In discrete action domains, we compute the log-sum-exp exactly by invoking the standard `tf.reduce_logsumexp()` (or `torch.logsumexp()`) functions provided by autodiff libraries. In continuous action tasks, CQL($\mathcal{H}$) uses importance sampling to compute this quantity, where in practice, we sampled **10** action samples at every state $\mathbf{s}$ from a uniform-at-random $\text{Unif}(\mathbf{a})$ and **10** action samples from the current policy, $\pi(\mathbf{a}|\mathbf{s})$ and used these alongside importance sampling to compute it as follows using $N = 10$ action samples:

$$\log \sum_{\mathbf{a}} \exp(Q(\mathbf{s}, \mathbf{a})) = \log \left( \frac{1}{2} \sum_{\mathbf{a}} \exp(Q(\mathbf{s}, \mathbf{a})) + \frac{1}{2} \sum_{\mathbf{a}} \exp(Q(\mathbf{s}, \mathbf{a})) \right)$$

$$= \log \left( \frac{1}{2} \mathbb{E}_{\mathbf{a} \sim \text{Unif}(\mathbf{a})} \left[ \frac{\exp(Q(\mathbf{s}, \mathbf{a}))}{\text{Unif}(\mathbf{a})} \right] + \frac{1}{2} \mathbb{E}_{\mathbf{a} \sim \pi(\mathbf{a}|\mathbf{s})} \left[ \frac{\exp(Q(\mathbf{s}, \mathbf{a}))}{\pi(\mathbf{a}|\mathbf{s})} \right] \right)$$

$$\approx \log \left( \frac{1}{2N} \sum_{\mathbf{a}_i \sim \text{Unif}(\mathbf{a})}^{N} \left[ \frac{\exp(Q(\mathbf{s}, \mathbf{a}_i))}{\text{Unif}(\mathbf{a})} \right] + \frac{1}{2N} \sum_{\mathbf{a}_i \sim \pi(\mathbf{a}|\mathbf{s})}^{N} \left[ \frac{\exp(Q(\mathbf{s}, \mathbf{a}_i))}{\pi(\mathbf{a}_i|\mathbf{s})} \right] \right).$$

On some domains, we observe that choosing higher values of $N$ is better, since it is more effective in covering the space of Q-values and prevents overestimation.

**Hyperparameters.** For the D4RL tasks, we built CQL on top of the implementation of SAC provided at: `https://github.com/vitchyr/rlkit/`. Our implementation mimicked the RLkit SAC algorithm implementation, with the exception of a smaller policy learning rate, which was chosen to be 3e-5 or 1e-4 for continuous control tasks except the gym-MuJoCo tasks, for which we used the default SAC policy learning rate of 3e-4. Following the convention set by D4RL [12], we report the normalized, smooth average undiscounted return over 4 seeds for in our results in Section 6. The other hyperparameters we evaluated on during our preliminary experiments, and might be helpful guidelines for using CQL are as follows:

- **Q-function learning rate.** We tried two learning rate values $[1e-4, 3e-4]$ for the Q-function. We didn't observe a significant difference in performance across these values. $3e-4$ which is the SAC default was chosen to be the default for CQL.

- **Policy learning rate.** We evaluated CQL with a policy learning rate in the range of $[3e-5, 1e-4, 3e-4$. We found $3e-5$ to almost uniformly attain good performance. While $1e-4$ seemed to be better on some of our experiments (such as hopper-medium-v0 and antmaze-medium-play-v0), but it performed badly with the real-human demonstration datasets, such as the Adroit tasks. As a result for the gym tasks, we chose a learning rate of $3e-4$ and in other cases we chose $3e-5$ as the learning rate.

- **Lagrange threshold $\tau$ and $\alpha$.** We ran our preliminary experiments with three values of threshold, $\tau = [2.0, 5.0, 10.0]$. However, we found that $\tau = 2.0$, led to a huge increase in the value of $\alpha$ (sometimes upto the order of millions), and as a result, highly underestimated Q-functions on all domains (sometimes upto the order of -1e6). On the datasets with human

demonstrations – the Franka Kitchen and Adroit domains, we found that $\tau = 5.0$ obtained lower-bounds on Q-values, whereas $\tau = 10.0$ was unable to prevent overestimation in Q-values in a number of cases and Q-values diverged to highly positive values (> 1e+6). For the MuJoCo domains, we observed that a static alpha $\alpha = 5.0$ gave rise to a stable curve of Q-values, without any dynamic selection rule for $\alpha$, and hence this threshold was chosen for these experiments. Note that none of these hyperparameter selections required any notion of onine evaluation, since these choices were made based on the trends of predicted Q-values on dataset state-actions pairs and based on how large the CQL loss is – if the CQL loss is too large, then the policy is going to pick unseen actions.

- **Number of gradient steps.** We evaluated our method on varying number of gradient steps. Since CQL generally uses a reduced policy learning rate (3e-5), we trained CQL methods for 1M gradient steps. For a number of the gym tasks (e.g., hopper and walker tasks), we find that CQL methods are trained in about 500k gradient steps, while for the halfcheetah tasks, they take 1M steps and are still improving. For instance on the halfcheetah-medium-expert dataset, we find that training for longer can give rise to better performance (e.g., $\approx 9k$ in return as compared to $\approx 7k$ as reported. Due to a lack of a proper valdiation error metric for offline Q-learning methods, deciding the number of gradient steps dynamically has been an open problem in offline RL. [34]. Different prior methods choose the number of steps differently. For our experiments, we used 1M gradient steps for all D4RL domains, and followed the convention from Agarwal et al. [3], to report returns after 5X gradient steps of training for the Atari results in Table 3.

- **Choice of Backup.** Instead of using an actor-critic version of CQL, we can instead also use an approximate max-backup for the Q-function in a continuous control setting. Similar to prior work [30], we sample 10 actions from the current policy at the next state $\mathbf{s}'$, called $\mathbf{a}_1, \cdots \mathbf{a}_{10} \sim \pi(\mathbf{a}'|\mathbf{s}')$ and generate the target values for the backup using the following equation: $r(\mathbf{s}, \mathbf{a}) + \gamma \max_{\mathbf{a}_1, \cdots, \mathbf{a}_{10}} Q(\mathbf{s}', \mathbf{a}')$. This is different from the standard actor-critic backup that performs an expectation of the Q-values $Q(\mathbf{s}', \mathbf{a}')$ at the next state under the policy's distribution $\pi(\mathbf{a}'|\mathbf{s}')$. In certain environments, such as in the Franka Kitchen and AntMaze domains using these backups performs better than the reported numbers with actor-critic algorithm.

Other hyperparameters, were kept identical to SAC on the D4RL tasks, including the twin Q-function trick, soft-target updates, etc. In the Atari domain, we based our implementation of CQL on top of the QR-DQN implementation provided by Agarwal et al. [3]. We did not tune any parameter from the QR-DQN implementation released with the official codebase with [3].

## G    Ablation Studies

In this section. we describe the experimental findings of some ablations for CQL. Specifically we aim to answer the following questions:

1. How does CQL($\mathcal{H}$) compare to CQL($\rho$), with $\rho = \hat{\pi}^{k-1}$, the previous policy?

2. How does the CQL variant which uses Equation 1 compare to CQL($\mathcal{H}$) variant in Equation 4, that results in a tighter lower-bound theoretically?

We start with question **(1)**. On three MuJoCo tasks from D4RL, we evaluate the performance of both CQL($\mathcal{H}$) and CQL($\rho$), as shown in Table 5. We observe that on these tasks, CQL($\mathcal{H}$) generally performs better than CQL($\rho$). However, when a sampled estimate of log-sum-exp of the Q-values becomes inaccurate due to high variance importance weights, especially in large action spaces, such as in the Adroit environments, we observe in Table 2 that CQL($\rho$) tends to perform better.

| Task Name | CQL($\mathcal{H}$) | CQL($\rho$) |
|---|---|---|
| halfcheetah-medium-expert | **7234.5** | 3995.6 |
| walker2d-mixed | **1227.2** | 812.7 |
| hopper-medium | **1866.1** | 1166.1 |

Table 5: Average return obtained by CQL($\mathcal{H}$), and CQL($\rho$) on three D4RL MuJoCo environments. Observe that on these environments, CQL($\mathcal{H}$) generally outperforms CQL($\rho$).

Next, we evaluate the answer to question **(2)**. On three MuJoCo tasks from D4RL, as shown in Table 6, we evaluate the performance of CQL($\mathcal{H}$), with and without the dataset maximization term in Equation 2. We observe that omitting this term generally seems to decrease performance, especially in cases when the dataset distribution is generated from a single policy, for example, hopper-medium.

| Task Name | CQL($\mathcal{H}$) | CQL($\mathcal{H}$) (w/ Equation 1) |
|---|---|---|
| hopper-medium-expert | 3628.4 | 3610.3 |
| hopper-mixed | **1563.2** | 864.6 |
| hopper-medium | **1866.1** | 1028.4 |

Table 6: Average return obtained by CQL($\mathcal{H}$) and CQL($\mathcal{H}$) without the dataset average Q-value maximization term. The latter formulation corresponds to Equation 1, which is void of the dataset Q-value maximization term. We show in Theorem 3.2 that Equation 1 results in a weaker lower-bound. In this experiment, we also observe that this approach is generally outperformed by CQL($\mathcal{H}$).