[Reviews · NeurIPS 2020]

Review 1

Summary and Contributions: This paper introduces conservative Q-learning (CQL) for offline reinforcment learning. The proposed algorithm attempts to reduce overestimation of Q-values at out of distribution actions by adding a regularization term to the usual Bellman backup that encourages minimizing Q values on actions sampled from some distribution other than the behavior. This is extended to a variant that maximized over this distribution (with additional regularization on the distribution). This is instantiated in a few different variants and shows strong performance on offline control benchmarks.

Strengths: 1. The paper presents a simple and novel aproach to combat overestimation in offline Q-learning by directly minimizing off-policy Q values instead of estimating the behavior density or the uncertainty in the Q function. This is a conceptually clever idea that can be a significant contribution to the offline RL community. 2. The proposed algorithm has good empirical performance on benchmark datasets. The empirical evaluation includes a large set of tasks and by using a previously defined benchmark provides useful comparison to recent work, often performing well in comparison.

Weaknesses: 1. The theoretical claims of producing lower bounds on Q values are not sufficient since there is no proof that the conservative Q values are anywhere near the true Q values. Just estimating Q = 0 for positive rewards could give the same result at theorems 3.1, 3.2, and 3.3. Clearly, the algorithm is doing something smarter than this, but the current analysis does not characterize what the algorithm is doing. The gap expanding result is likely the strongest of the four theorems, but without doing the work to connect this back to why this will actually help performance it is still difficult to judge. Moreover, no comparison is made to the overestimation that would happen without the proposed algorithmic change. 2. Evaluating the statements of the theorems more closely they seem to have errors and are not clearly non-vacuous. For example, in Theorems 3.1 and 3.2 the last term is a matrix multiplied by a scalar and the inequality is meant to be a scalar, so some term is clearly missing. In Theorem 3.1, I think the second term is meand to be multiplying the I - gamma P inverse matrix by a vector of density ratios and then taking the s,a element, but the notation is especially confusing since bold s and a are used for indices of the vecot being multiplied and the resulting product. Theorem 3.1 assumes that the support of mu is contained in pi_beta, but then goes on to claim that the result holds for all s,a. This can't be the case since the second term in the bound is not even properly defined at a state,action pair with pi_beta =0. I think an assumption about pi_beta having full coverage or limiting the scope of s,a pairs is necessary. Additionally, the values for alpha required to make the Theorem actually yield prevention of overestimation are so large (large multiples of the maximum Q value), that it is not clear if the theorems are impactful. 3. One smaller issue if that the claim made a few times, for example on lines 82-83, that "offline RL does not suffer from state distribution shift" seems to be wrong or at least imprecise. If the learned policy (that maximizes the learned Q function) is different from the behavior policy, then at test time the induced state distribution will be different from the data distribution over states. States may be sampled from the behavior distribution at train time, but at test time this will be far from the case and this mismatch could cause serious problems. In this way, the state distribution shift is indeed a major issue.

Correctness: As stated above, the empirical method seems sound, but the theorem statements seem to have at least some surface level inconsistencies and also lack completeness.

Clarity: The paper is fairly clear with a few notable exceptions. 1. As mentioned above, the theorem statements have poor notation and are quite opaque. Improved notation and more intuitive explanation of the terms in the bounds would be helpful. 2. In general too much space is taken up by the theoretical section that does not lay out any clear benefits for the algorithm relative to related approaches. Either more comparative theoretical analysis or expanded experiments like Appendix B in the main paper could greatly improve clarity.

Relation to Prior Work: The connections to prior work are well cited throughout. A few things could be improved: 1. Connections to the theoretical literature on batch RL like [1] and [2] should be included. 2. Experiments are a strongpoint of this paper, but just to nitpick it would be nice to include comparisons to ABM [3], BCQ [4], and AWR [5] in the experiments section. Especially since AWR and ABM also do not require estimating the behavior policy, which is one of the main claimed advantages of BCQ. [1] Munos, Rémi, and Csaba Szepesvári. "Finite-time bounds for fitted value iteration." *Journal of Machine Learning Research* 9.May (2008): 815-857. [2] Chen, Jinglin, and Nan Jiang. "Information-theoretic considerations in batch reinforcement learning." *arXiv preprint arXiv:1905.00360* (2019). [3] Siegel, Noah Y., et al. "Keep doing what worked: Behavioral modelling priors for offline reinforcement learning." *arXiv preprint arXiv:2002.08396* (2020). [4] Fujimoto, Scott, David Meger, and Doina Precup. "Off-policy deep reinforcement learning without exploration." *International Conference on Machine Learning*. 2019. [5] Peng, Xue Bin, et al. "Advantage-weighted regression: Simple and scalable off-policy reinforcement learning." *arXiv preprint arXiv:1910.00177* (2019).

Reproducibility: Yes

Additional Feedback: As explained above, the empirical results are strong and the proposed algorithm is elegant, but I think the theoretical analysis needs to be substantially refined before the paper is ready to be accepted. Minor things and typos: 1. The analysis seems to assume finite state and action spaces but I never saw this explicitly stated. 2. Table 3 caption should be "setting (2)". 3. Lines 528 and 543 in Appendix B have typos. 4. The first paragraph of the broader impact section is largely summarizing the paper, not explaining broader impacts. ----------------------------------------------------------------------------------------------- ----------------------------------------------------------------------------------------------- Post-rebuttal update: The rebuttal does a good job of addressing my central concerns about the theoretical section of the paper. First, the addition of a policy improvement result helps to assuage my concern that theorems 3.1-3.3 were potentially vacuous. Second, the authors claim to fix all of the inconsistencies in the statements of the theorems and to add the necessary coverage assumptions to ensure that the results are meaningful and correct. And third, the authors claim to shift the emphasis of the paper more toward the empirical results in offline deep RL and away from the theoretical results, which may provide useful motivation, but are not the main contribution of this paper. Unfortunately, without being able to review all of the claimed changes in detail, I'm hesitant to raise my score to a strong accept. I have increased my score to 6 and decreased my confidence to 3 to reflect the uncertainty about the unseen changes.


Review 2

Summary and Contributions: This paper introduces a new update rule for the Q-function that regularizes the Q-values by minimizing its values under an appropriately chosen auxiliary distribution over state-action pairs. The authors show that these update rules asymptotically result in a lower bound to the value of the policy obtained from exact policy evaluation. The paper then uses this conservative Q-function in learning algorithms to obtain The main contribution is the conservative Q-Learning algorithm derived from this update rule. The authors show that the policy updates derived in this way are conservative, in the sense that at each iteration the policy is optimized against a lower bound on its value. Further, they show that their method is more robust against out of distribution errors induced by the behavior policy.

Strengths: The paper investigated offline reinforcement learning, which is of increasing interest to the NeurIPS community. Making RL work reliably on offline datasets could make RL significantly more applicable to real-world problems. Moreover, the presented approach is novel as regularizing the Q function to be pessimistic for OOD actions has to my knowledge not been investigated for offline RL before. The approach in the paper is motivated by theoretical results in discrete MDPs, where the paper shows that the estimated Q/value function lower-bounds the true value function. In particular, by how much the method lower-bounds the value depends on the empirical behavior distribution pi_beta, where actions that have been evaluated infrequently lead to a larger lower-bound. This theoretical foundation motivates the approach jointly with function approximation.

Weaknesses: While the theoretical motivation of the paper focuses on constructing lower bounds, there is no discussion about by how much and whether this is reasonable. In fact, both (1) and (2) are unbounded optimization problems for actions that are not in the empirical distribution, which is also reflected by divide by zero (-inf lower bound) in both Theorem 3.1 and 3.2 when supp mu > supp pi_beta, I.e., for actions that are not part of the dataset. This will always be the case in continuous settings, which hints at that regularizing the Q function should be of critical importance for this method to work well. There is relatively little discussion how this approach relates to other methods that regularize the policy directly to be close the the (estimated) behavior policy. It seems to me that while other approaches rely on estimating the behavior policy for regularization, this approach relies on generalization properties of the Q-function instead. It's not obvious that this is necessarily better. A main weakness of the approach lies in the alpha factor that trades of a squared loss in Q values relative to a linear difference of Q-values. These two will have fundamentally different sizes in general and, as the average magnitude of Q-values is highly problem-specific, it seems difficult to select an alpha that will work well in general. The main paper ignores this completely and only states that this is auto-tuned through the Lagrangian. However, the solution in the appendix boils down to "I'll turn the soft constraint into a hard constraint", which does not avoid the difference between squared and non-squared values. This is also evident from the discussion about the difficulty of selecting an appropriate value for the constraint tau in section F. The empirical evaluation is broad in terms of different environments, but does not provide any statistics beyond the mean (and even that only averaged over four seeds). Providing statistics in form of standard error or population standard deviation is essential to draw significant conclusions from the tables. Normalizing the returns also makes it difficult to compare to published results in online learning.

Correctness: The theoretical derivations seem correct. The empirical methodology of comparing means rather than providing proper statistics is flawed (see weakness section).

Clarity: Overall the paper is well-written and easy to follow. Some more discussion would be appreciated at times. I would expect that the preliminaries are impossible to parse for a reader that is not familiar with RL. While I appreciate space constraints, at least briefly defining value and Q functions in terms of the returns and explaining/defining the discounted state distribution d would be desirable. Right now the value function is not defined anywhere.

Relation to Prior Work: While I think most of the related work is covered, providing more details about the differences between the methods in the related work section would be desirable.

Reproducibility: Yes

Additional Feedback: Questions to authors: Is it obvious that (1) and (2) will necessarily converge to a fixed point (ignoring actions that are not part of D)? Could you explain how methods that regularize towards the BC policy can perform worse than the BC policy? It seems to me that I can always pick the regularization strong enough to effectively enforce equivalence to the BC policy, so how come this is not the case in Table 1/2? General comments: At the end of Thm 3.1 I would assume any alpha >= 0 (rather than >0) guarantees Qhat <= Qpi Avoid \textbf{} inside the text (e.g., in the introduction). Use \emph instead. Line 111 and 114: ‘behavior policy’ should be replaced by ‘empirical behavior policy’, i.e. no approximation of the true behavior policy is required. For conditional probabilities consistently use $pi(a \mid s)$ rather than $pi(a|s)$. I would prefer for the Theorems to be self-contained, so that, e.g., the concentration property of Bhat and B, shows up in the text of the theorem, rather than in the main body just before. Post-rebuttal: ------------------ The rebuttal is well-written and addresses several of the concerns raised by the reviews.


Review 3

Summary and Contributions: The authors tackle the problem of learning a new RL policy from offline data. This problem is hard because the algorithms tend to overestimate outcomes in states that are rarely (or not) encountered in the dataset. The first idea proposed in the submission consist in penalizing large Q-value estimates for the target policy. This allows to produce underestimates of the target policy value, but this underestimation is too conservative. The second idea consists in substracting the value of the baseline policy. This is also proven to produce underestimations of the target policy value. The authors analyze both theoretically and empirically this idea.

Strengths: - I believe that batch RL has been understudied for a very long time and I am happy to observe its regain of popularity since 2-3 years. - I really like the 2 successive ideas to obtain underestimates of the target policy value. They are smart because, contrarily to most existing methods they do not need any form of state visit density. - The empirical results are extensive and quite good.

Weaknesses: W1: There is no theoretical result on the near optimality, or the policy improvement guarantees offered by the algorithms. W2: Correctness: I have several several major concerns on correctness. See below C1, C3, C4, and C5. W3: Relation to prior work is insufficient. See below R1, R2, R3, R4, and R5. W4: Experiments: I would like to see the baseline/dataset performance as a reference in Tables 1-2. Also, the fact that behavioral cloning is the second best performing algorithm, tells that the other algorithms did not work, which is surprising since they were published in part because they were better than behavioral cloning. Is there an explanation for this?

Correctness: Herebelow, I enumerate the correctness concerns I have: C1: the empirical behaviour policy is formally described lines 65-66, it is said that \mu must satisfy some condition with respect to it line 631, but it is later claimed line 245-246 that the empirical behaviour policy does not need to be estimated. Also, it is unclear what the empirical behaviour policy would be in infinite MDPs. C2: lines 82-84: I believe I understand what the authors meant but this claim seems wrong to me: the Q-function training is still biased by the distribution with function approximation. (minor) C3: the way how \alpha is set is quite fuzzy to me. First of, alpha seems to be very large, even infinite if we consider that \mu could equal 0 in theorem 3.1, of that \pi could equal ^\pi_\beta in Theorem 3.2. As long as we only consider a single policy to evaluate (as in Section 3.1), we may compute it, but when we try to optimize \pi with respect to the objective, then the fact that \alpha is dependent on \pi is a big issue. I did not see an explanation of that anywhere.

Clarity: (still correctness concern, I was limited by the size of its box) C4: Equation between lines 847 and 848: there is no guarantee that (s',a') is in D? So |D(s',a')| might be equal to 0? And therefore the conclusion may not hold? Also, the fact that the squareroot of D(s,a) is dropped is disturbing to me. It means that your theoretical results do not improve with more data, it is probably much weaker than Petrik's theorem (see R2), it somehow indicates that the constants must be gigantic. Finally, Assumption D.1 generally holds only when D(s,a) is sufficiently large (say 30). So, you might as well replace Assumption D.1 with |r-^r|<e_r and |t-^t|<e_t. In this case, Section D.3 amounts to classical results in eR_max/(1-\gamma) for a single policy evaluation. But further, and more importantly, it has been proven that under this assumption, dynamic programming on the MDP estimate produces a policy that is near-optimal with a value approximation in eR_max/(1-\gamma)^2 (see Theorem 8 of [Petrik2008]). The subsequent theoretical findings of the submission are much weaker than that. C5: first equation after line 626: I think that there is a mistake here because the (s,a) pair is not the one as an argument of Q. Later, when the min over (s,a) couples is taken, the result gets corrected , but the theorems itself are wrongly formulated. Clarity evaluation: The paper is well written in general. Here is a couple of things I did not fully understand, and that should maybe be rewritten (these are minor remarks): - lines 75-77 - Theorem 3.4: what is \mu_k here ? Also it seems to me that the result would be easier to interpret had the expectations on the same distributions had been kept in the same side of the inequality. If I understand it well, it means that the underestimation of Q is larger in the baseline distribution than in the learnt one.

Relation to Prior Work: Despite the extensive amount of references, I believe that the submission misses several critical references to position their work with respect to prior work. In general, the ideas and theoretical results are never compared to the literature. There is a related work section, but the discussion focuses on pros and cons rather on the ideas and results. Also, the citations seem biased: only 11 out of 48 citations do not include an author that is currently working in Alphabet (basically Google, Deepmind or Google Brain). I understand that this corporation generates a high volume of very good contributions to the machine learning community, but [NeurIPS2019] statistics report that only 170/1428 < 12% were produced by Alphabet, [ICML2017] reports 6.3%, and [ICML2020] reports ~14%. Please, find below a list of papers and places in the submission where I believe that the reference were missing: R1: The name of the algorithm and the setting of an improvement parameter \alpha, constraining the amount of allowed policy improvement inevitably recalls Conservative Policy Iteration [Kakade2002]. It seems necessary to me to make a comparison. R2: Lines 136-137: since we are in a finite state setting, how do Theorems 3.1 and 3.2 compare to Theorem 8 of [Petrik2016]. They seem weaker to me. R3: [Laroche2019] uses [Petrik2018] theorem to construct several algorithms, later empirically improved by [Nadjahi2019] and [Dias2020], and it would be informative to have some kind of comparison to at least one of this thread of research. Similarly, 2 other important threads of research has been left out of the paper: R4: Robust MDPs: [Iyengar2005] and [Nilim2005]. These algorithms have been shown several times to be too conservative. It would be interesting to hear some arguments to understand why CQL would be less conservative. R5: High confidence policy improvement of [Thomas2015a] and [Thomas2015b]. The importance sampling offline evaluation is briefly discussed, but not the Batch RL algorithms based on them. These are other algorithms that would deserve a place in the benchmark in my opinion. [Dias2020] Simão, T. D., Laroche, R., & Combes, R. T. D. (2020). Safe Policy Improvement with an Estimated Baseline Policy. AAMAS, 2020. [ICML2017] https://medium.com/@karpathy/icml-accepted-papers-institution-stats-bad8d2943f5d [ICML2020] https://medium.com/criteo-labs/icml-2020-comprehensive-analysis-of-authors-organizations-and-countries-c4d1bb847fde [Iyengar2005] Iyengar, G. N. Robust dynamic programming. Mathematics of Operations Research, 30(2):257–280, 2005. [Laroche2019] Laroche, R., Trichelair, P., & Tachet des Combes, R. T. (2019, May). Safe policy improvement with baseline bootstrapping. In International Conference on Machine Learning (pp. 3652-3661). [Nadjahi2019] Nadjahi, K., Laroche, R., & des Combes, R. T. (2019, September). Safe policy improvement with soft baseline bootstrapping. In Joint European Conference on Machine Learning and Knowledge Discovery in Databases (pp. 53-68). Springer, Cham. [NeurIPS2019] https://medium.com/@dcharrezt/neurips-2019-stats-c91346d31c8f [Nilim2005] Nilim, A. and El Ghaoui, L. Robust control of Markov decision processes with uncertain transition matrices. Operations Research, 53(5):780–798, 2005. [Petrik2016] Petrik, M., Ghavamzadeh, M., & Chow, Y. (2016). Safe policy improvement by minimizing robust baseline regret. In Advances in Neural Information Processing Systems (pp. 2298-2306). [Thomas2015a] Thomas, P. S., Theocharous, G., & Ghavamzadeh, M. (2015, February). High-confidence off-policy evaluation. In Twenty-Ninth AAAI Conference on Artificial Intelligence. [Thomas2015b] Thomas, P., Theocharous, G., & Ghavamzadeh, M. (2015, June). High confidence policy improvement. In International Conference on Machine Learning (pp. 2380-2388).

Reproducibility: Yes

Additional Feedback: As stated before, I have major concerns regarding positioning to prior work and correctness. For these reasons I recommend to reject the paper. Typos (minor): - line160: first - equations inside lines 649-650: missing parenthesis. __________________________________________________ __________________________________________________ __________________________________________________ Post rebuttal comments: I have to say that I have been very impressed by the rebuttal. The authors address all my main points with compelling promises. Nevertheless, I have noticed that C3 has not been addressed in the rebuttal (C3 was not the one related to the tightened bounds with counts, it was C4). I will consider this as a honest mistake, and judging by the efficiency to address my other concerns, I hope that the authors will improve their paper with regard to this concern. Given the fact that I cannot check all their rebuttal claims and I'm worried about the existence of other mistakes that I missed, I'm only changing my score from 3 to 6, which should be enough for acceptation. I deeply regret that the theory was not polished and that their positioning was this much biased before submission, because, if all the promises are delivered, the paper would probably have deserved special attention (spotlight or even oral).


Review 4

Summary and Contributions: The paper "Conservative Q-Learning for Offline Reinforcement Learning" proposes a novel approach for fully offline Q-learning. One major problem in offline Q-learning is that the available samples only contain actions for a small subset of possible actions making standard Q-learning hard. The technical insight in the paper is to update the Q function in two ways. Firstly, a weighted sum of a Bellman residual and the Q function is minimized resulting in a lower bound for the Q-function. Secondly, when adding a third term to maximize the Q function under a policy derived from the current Q function leads to a lower bound for the state value function V. These technical insights into how to regularize using Q functions, together with entropy regularization, provide algorithms that outperform state-of-the-art offline RL algorithms in a wide variety of tasks. The paper provides proofs for the lower bounds and further analysis.

Strengths: - Key technical insights into how to construct a lower bound without making it too loose - Analytical proofs that the proposed Q estimates lower bound the true Q and V functions - Theoretical analysis of the behavior of the proposed techniques - Strong empirical results against state-of-the-art methods in a wide variety of benchmark tasks - Offline RL is important in many different domains. For example, in robotics data can be gathered during operation and online RL is not feasible due to operating costs. Furthermore, the proposed techniques will with high probability find application in other RL contexts.

Weaknesses: - Intuitive explanation of how the lower bounds work would be beneficial - The discussion on regularizing using the minimimums and maximums of Q functions and then later using (entropy or similar) regularization is convolved. It should be made clear that there are two kinds of regularization and that the (entropy or similar) regularization is needed to get a policy that covers all actions and which can then be used in the Q function based regularization proposed in the paper.

Correctness: I did not find errors in the claims, methods, or the empirical methodology.

Clarity: The paper is in principle well written except for a couple of points. A more intuitive explanation of how the lower bounds work would be beneficial. In addition, now it is unclear what \mu(a|s) denotes. The relationship of \mu and \pi_{\beta} is crucial in the Equations and Theorems. \pi_{\beta} is explained in the beginning of Section 2 but \mu should be also explained there. Reading the paper and appendix further clarifies the role of \mu(a|s) but this should be made clear before using \mu(a|s) in Equation 1. In particular, it would be beneficial to mention that \mu(a|s) covers all actions since it is derived from an entropy regularized form of the Q function. Details: Line 63: "action-conditional" is a non-standard term, please define. Line 160: "Frist" -> "First" In Equation 2, there is E_{s,a,s' ~ D} on the RHS but in Equation 1 there is E_{s,a ~ D} on the RHS.

Relation to Prior Work: Overall the discussion of related work is good. More details on the differences to [22, 46] should be added.

Reproducibility: Yes

Additional Feedback: Author rebuttal: Thank you for the rebuttal. For practical implementation I hope the final version clarifies things such as that the policy needs to be non-zero for all actions (according to the rebuttal this will be added to the Theorems to make them correct).

[Author Response · NeurIPS 2020]

We thank the reviewers for their constructive feedback. **We have reformulated the theoretical statements to position them with previous work, clarified notation and surface-level inconsistencies in theoretical statements, added experiment baselines, and performed an additional comparison with SPIBB (Laroche et al. 2019).** The main contribution of our work is a *practical* offline deep RL algorithm, CQL, which attains strong results and outperforms prior methods by a large margin on a wide range of tasks. We have revised the paper to frame the theoretical results as providing motivation for our approach rather than a primary contribution.

**Additional experiments & baselines:** Fu et al. (2020) have added results for AWR, BCQ, REM and AlgaeDICE in the D4RL paper, which we now include in Tables 1 & 2. We have added an explicit mention that the baseline numbers are from Fu et al. 2020 **[R2, (R3, W4)]** and will add variance measurements **[R2]**. CQL outperforms AWR, BCQ, REM and AlgaeDICE **(R1)** in **26/29** tasks, often by a large margin. **(R3)** We also compared CQL to **SPIBB** on the Helicopter task from Laroche et al. 2019. With a dataset of size 10k, CQL outperforms SPIBB by attaining **4.11** mean return whereas SPIBB (w/ best $N_\wedge$) attains **3.22** return and soft-SPIBB (w/ best $\epsilon$) attains **3.65** return.

**R1/[R3, W1]: Policy improvement result, what is CQL doing.** Based on R1/R3's requests, we have added a new theorem for policy improvement property of CQL that is more consistent with prior work. Similar to Thm. 1 in Laroche et al. 2019, we show that the CQL updates (Eqn. 2) converge to the optimal policy of a (empirical) penalized RL objective, i.e., $\pi_{\mathrm{CQL}} = \arg\max_\pi \hat{J}_\mathcal{D}(\pi) - \alpha\mathbb{E}_{\mathbf{s}\sim\hat{d}^\pi}[D_{\mathrm{CQL}}(\pi,\pi_\beta)](1-\gamma)^{-1}$, where $\hat{J}_\mathcal{D}(\pi)$ is an empirical estimate of expected discounted return and $D_{\mathrm{CQL}}$, defined in lines 649-650, captures a notion of action distribution shift. Then, using the terminology from Laroche et al. 2019, we show that $\pi_{\mathrm{CQL}}$ is a $\zeta$-approximate safe policy improvement over $\pi_\beta$ (i.e. $J(\pi_{\mathrm{CQL}}) \geq J(\pi_\beta) - \zeta$) with high probability $1-\delta$, where

$$\zeta = \frac{\gamma C\sqrt{\log(1/\delta)}}{(1-\gamma)^2}\mathbb{E}_{\mathbf{s}\sim\hat{d}^{\pi_{\mathrm{CQL}}}(\mathbf{s})}\left[\frac{\sqrt{|\mathcal{A}|}}{\sqrt{|\mathcal{D}(\mathbf{s})|}}\sqrt{D_{\mathrm{CQL}}(\pi_{\mathrm{CQL}},\pi_\beta)(\mathbf{s})+1}\right] - \overbrace{\left(\hat{J}_\mathcal{D}(\pi_{\mathrm{CQL}}) - \hat{J}_\mathcal{D}(\pi_\beta)\right)}^{\geq\alpha\mathbb{E}_{\mathbf{s}\sim\hat{d}^{\pi_{\mathrm{CQL}}}(\mathbf{s})}[D_{\mathrm{CQL}}(\pi_{\mathrm{CQL}},\pi_\beta)(\mathbf{s})](1-\gamma)^{-1}} .$$

This follows from applying and extending the tools in Achiam et al. 2017. Note the similarity with Thm. 2 in (Laroche et al. 2019) and that we avoid the $\infty$-norm term in (Petrik et al. 2016). Intuitively, this indicates that the return of $\pi_{\mathrm{CQL}}$ is higher than the behavior policy $\pi_\beta$ w.h.p. when the empirical policy improvement (i.e., the $\alpha$ term) exceeds the sampling error, which diminishes to 0 as the dataset size (i.e., $|\mathcal{D}(\mathbf{s})|$) grows.

**R2: Behavior policy estimation.** Since our submission, Nair et al. 2020 and Ghasemipour et al. 2020 have discussed the difficulty of behavior policy estimation at length. We will add extended discussion of this point in the paper.

**R1/(R3, W2): Clarifying notation and resolving surface level inconsistencies.** We have resolved the notational confusion and inconsistencies in the results. We discuss the main changes briefly:
- **(Also R3, C5)** We now explicitly indicate dimensions of vectors, matrices and scalars. Briefly, $[\mu(\mathbf{a}|\mathbf{s})/\pi_\beta(\mathbf{a}|\mathbf{s})]$ denotes a vector of size $|\mathcal{S}||\mathcal{A}|$ equal to elementwise ratio of $\mu$ and $\pi_\beta$ and this is multiplied to the matrix $(I-\gamma P^\pi)^{-1}$.
- **(Also R3, C1)** $\pi_\beta(\mathbf{a}|\mathbf{s})=0$: We have added a discussion to indicate that when $\pi_\beta(\mathbf{a}|\mathbf{s})$ is zero in the tabular setting, the learned Q-values, $\hat{Q}(\mathbf{s},\mathbf{a})$ can be $-\infty$. For Thm 3.1 and 3.2, we have edited these to include the assumption that $\pi_\beta(\mathbf{a}|\mathbf{s})$ has non-zero density on all actions to prevent $-\infty$ values, and clarified that this result holds only for $\mathbf{s}\in\mathcal{D}$.

**R1/[R3, C3]: Tightened bounds with counts.** For simplicity, we had omitted the count terms from the main text, instead deferring discussion to App. D.3. However, as the reviewers point out, they are necessary. We now include the count terms that scale inversely proportional to the square root of $|\mathcal{D}(\mathbf{s},\mathbf{a})|$ in the main text. As expected, the sampling error and theoretically safe value of $\alpha$ both decrease as the dataset size increases. In practice, the theoretical value of $\alpha$ is overly-conservative, so as done in previous work, we use a smaller $\alpha$ rather than this theoretical value.

**(R3, W3): Relation to prior work.** We have added and positioned our ideas with respect to the suggested references on robust MDPs, CPI, safe PI, high confidence PI, batch RL **(Also R1)**, including **12** references from non-Alphabet authors. We suspect that CQL outperforms robust MDPs empirically due to the approximations necessary to adapt robust MDPs to a practical algorithm. We have also added an empirical comparison to SPIBB.

**R1/[R3, C2]: Offline RL and state distribution shift.** We have clarified these statements (e.g., lines 83-84) to say that *training procedure* for a Q-learning algorithm queries out-of-distribution actions which can lead to divergence.

**(R3, C4): Lines 847-848: no guarantee** $(\mathbf{s}',\mathbf{a}')\in\mathcal{D}$**?.** To clarify, the proof uses concentration of $\hat{T}(\cdot|\mathbf{s},\mathbf{a})$ which depends on the counts $|\mathcal{D}(\mathbf{s},\mathbf{a})|$. So while the inequalities involve $(\mathbf{s}',\mathbf{a}')$ through $\mathbb{E}[\hat{Q}^k(\mathbf{s}',\mathbf{a}')]$, we bound that term by $2R_{\max}(1-\gamma)^{-1}$ avoiding concerns over the size of $|\mathcal{D}(\mathbf{s}',\mathbf{a}')|$. We explicitly mention this now.

**R1: Intuition for gap expanding.** We have elaborated on this in the paper now. By increasing the difference between Q-values at in-distribution actions ($\pi_\beta$) and under learned policy ($\mu_k$), CQL constrains the resulting policy to lie in the support of $\mathcal{D}$. This controls the $D_{\mathrm{CQL}}(\pi,\pi_\beta)$ term that appears in the sampling error component of $\zeta$ in our $\zeta$-safe policy improvement result above. Empirically, this also makes CQL more robust to Q-function approx. error that makes OOD actions appear more optimal and results in better performance than policy constraint methods (App. B, Fig. 2).

[Meta-Review · NeurIPS 2020]

All the reviewers were positively impressed by the rebuttal provided by the authors, which clarified many of their concerns. Their updated scores and the final decision to propose acceptance for the paper are based on the requirement that the authors will significantly change the paper integrating the insights presented in the rebuttal as well as clarifications of the few remaining points that have not be addressed yet (see R3).